# Global stable-isotope tracing metabolomics reveals system-wide metabolic alternations in aging *Drosophila*

Ruohong Wang[1,2], Yandong Yin[1], Jingshu Li[1,2], Hongmiao Wang[1,2], Wanting Lv[1,2], Yang Gao[1,2], Tangci Wang[1], Yedan Zhong[1], Zhiwei Zhou [1], Yuping Cai[1], Xiaoyang Su [3,4], Nan Liu[1✉] & Zheng-Jiang Zhu [1✉]

System-wide metabolic homeostasis is crucial for maintaining physiological functions of living organisms. Stable-isotope tracing metabolomics allows to unravel metabolic activity quantitatively by measuring the isotopically labeled metabolites, but has been largely restricted by coverage. Delineating system-wide metabolic homeostasis at the whole-organism level remains challenging. Here, we develop a global isotope tracing metabolomics technology to measure labeled metabolites with a metabolome-wide coverage. Using *Drosophila* as an aging model organism, we probe the in vivo tracing kinetics with quantitative information on labeling patterns, extents and rates on a metabolome-wide scale. We curate a system-wide metabolic network to characterize metabolic homeostasis and disclose a system-wide loss of metabolic coordinations that impacts both intra- and inter-tissue metabolic homeostasis significantly during *Drosophila* aging. Importantly, we reveal an unappreciated metabolic diversion from glycolysis to serine metabolism and purine metabolism as *Drosophila* aging. The developed technology facilitates a system-level understanding of metabolic regulation in living organisms.

---

[1] Interdisciplinary Research Center on Biology and Chemistry, Shanghai Institute of Organic Chemistry, Chinese Academy of Sciences, 200032 Shanghai, People's Republic of China. [2] University of Chinese Academy of Sciences, 100049 Beijing, People's Republic of China. [3] Metabolomics Shared Resource, Rutgers Cancer Institute of New Jersey, Rutgers University, New Brunswick, NJ 08901, USA. [4] Division of Endocrinology, Department of Medicine, Robert Wood Johnson Medical School, Rutgers University, New Brunswick, NJ 08901, USA. ✉email: liunan@sioc.ac.cn; jiangzhu@sioc.ac.cn

System-wide metabolic homeostasis and coordination are of central importance to maintaining physiological functions for living organisms[1,2]. Disturbance to metabolic homeostasis causes cellular malfunctions and several major human diseases[3,4]. In the past two decades, untargeted metabolomics has been developed to measure differences in metabolite concentrations between biological conditions, aiming to provide system-wide characterizations of metabolic homeostasis in living systems[5–7]. However, changes in metabolite concentrations do not readily imply alterations in metabolic pathway activities, since metabolite levels measured are the converged results of both production and consumption from multiple metabolic reactions[8–12]. Alternatively, stable-isotope tracing-based metabolomics administers isotopic tracers such as [13]C-glucose and [13]C-glutamine into living systems, and tracks the isotopically labeled metabolite products generated by cellular metabolism[9,11]. By quantifying labeling patterns and fractions of the labeled metabolites, this technology enables unraveling metabolic activity at both the cellular and organismal levels[13–18]. Yet, many isotope tracing studies were largely restricted by targeting only a limited number of metabolites in specific pathways, and it is difficult to delineate the system-wide metabolic homeostasis with a high metabolite coverage[11,19]. Recently, combination of untargeted metabolomics and isotope tracing analysis, such as X[13]CMS[20,21], geoRge[22], and others[23–29], has been developed to comprehensively characterize isotope labeled metabolites in an untargeted manner, and provides possibilities to discover new metabolic transformations and pathways. Owing to the complexity of isotope labeling metabolomics data, however, untargeted isotope tracing metabolomics usually has relatively low coverages in the detection of labeled metabolites and lacks quantitative information such as labeling rates on a large scale. Global tracking of the isotopically labeled metabolites remains the bottleneck to delineate metabolic activities with a metabolome-wide coverage. More challenging is to calculate absolute fluxes for hundreds of metabolites in a pathways-intertwined metabolic network, wherein complicated mathematical frameworks and algorithms are usually required. These challenges are particularly prominent in mammalian systems, which impedes a large-scale quantitative investigation of system-wide metabolic homeostasis[10,11,19].

Disruption of metabolic homeostasis has been recognized as a hallmark of aging[4,30–32]. Increasing studies have demonstrated the age-dependent and system-wide dysregulation in metabolic pathways in different organisms. Various metabolic interventions have been proven to be beneficial to extend lifespans in different model organisms[33–36]. Among them, *Drosophila melanogaster* (hereafter *Drosophila*) is an important model organism for metabolism research[37,38], and has emerged as a model system to investigate metabolic alternations and organismal homeostasis during aging[34,39,40]. For instance, in *Drosophila*, levels of intermediate metabolites in the glycolysis pathway progressively declined with aging, and boosting glycolysis was demonstrated to extend lifespan[41]. Instead, S-adenosylmethionine (SAM) levels are increased during *Drosophila* aging, while enhancing SAM catabolism extends the lifespan[35]. Despite increasing aging-related metabolites and metabolic pathways are uncovered, these studies were independently investigated, and the underlying mechanism by which coordination of metabolic activities impacts aging is not elucidated due to the lack of appropriate technologies. In addition, quantitative investigations of metabolic activities during aging on a metabolome-wide coverage has not been achieved yet, which hinders the delineation of the complex milieu of metabolic coordination and the system-level understanding of metabolic regulation of aging and longevity.

In this study, we first developed an untargeted isotope tracing metabolomics technology, namely MetTracer, to trace the stable-isotope labeled metabolites globally. This technology leveraged the advantages of untargeted metabolomics and targeted extraction to track the isotopically labeled metabolites in living organisms with a metabolome-wide coverage. MetTracer enabled to simultaneously quantify the labeling patterns and extents of several hundreds of metabolites in one experiment. Performances of MetTracer were benchmarked and validated with other existing tools. We further employed *Drosophila* as a model organism for untargeted in vivo isotope tracing metabolomics and quantified the in vivo labeling rates and extents of hundreds of labeled metabolites. We demonstrated that MetTracer supported quantitative comparisons of metabolite labeling extents across different conditions and discovered a system-wide loss of metabolic coordination that impacted both intra- and inter-tissue metabolic homeostasis significantly in aging *Drosophila*. In particular, we discovered a metabolic rewiring model during aging, wherein glucose was metabolically channeled to serine metabolism and purine metabolism from glycolysis. In summary, global in vivo isotope tracing metabolomics enabled delineation of metabolic activities with a metabolome-wide coverage and unraveled the system-wide loss of metabolic homeostasis during aging.

## Results

**Global stable-isotope tracing metabolomics**. Stable-isotope tracing technology is significantly restricted by the analysis coverage of labeled metabolites. In this work, we developed a global stable-isotope tracing metabolomics workflow, namely MetTracer, to trace the isotopically labeled metabolites in a metabolome-wide coverage (Fig. 1a). MetTracer leveraged high coverage of untargeted metabolomics and high accuracy of targeted extraction. In brief, both unlabeled and labeled samples were analyzed using liquid chromatography–mass spectrometry (LC–MS)-based untargeted metabolomics. Metabolite annotation was first performed in unlabeled samples by matching experimental MS2 spectra against the standard spectral libraries and/or using bioinformatics tools (Supplementary Data 1–3). With annotated metabolites, MetTracer performed targeted extraction of all possible isotopologues with high accuracy through three major steps (Fig. 1b and Supplementary Fig. 1): (1) generation of a targeted list for isotopologues; (2) extraction of isotopologue peaks, and (3) isotopologue correction and quantification.

As a proof-of-concept, we analyzed 293T cell samples labeled with a mixture of tracers ([U-[13]C]-glucose, [U-[13]C]-glutamine and [U-[13]C]-acetate) using a time-of-flight (TOF) mass spectrometer. Specifically, 1347 metabolites were putatively annotated (215, 219, and 913 metabolites with MSI levels 1, 2, and 3, respectively). Then, the theoretical $m/z$ values for 12,020 possible [13]C-isotopologues (M0-Mn) were calculated from the formulas of 1347 metabolites. MetTracer performed targeted extraction of all possible isotopologues and successfully extracted a total of 10,663 isotopologues (88.7%) from 1203 metabolites (89.3%) (Supplementary Fig. 2a, b), which ensured the high-coverage tracking of labeled metabolites. Finally, MetTracer determined the labeled fraction for each isotopologue with the criterion of labeled fraction larger than 2% in >50% samples. As a result, MetTracer identified a total of 830 [13]C-labeled metabolites and 1725 [13]C-labeled isotopologues, which covered 66 metabolic pathways (Fig. 1c, d). Further benchmark analyses demonstrated that MetTracer improved the coverage of the labeled metabolites substantially compared to other tools such as X[13]CMS, El-MAVEN, and geoRge (Fig. 1d, Supplementary Table 1, Supplementary Data 4). Next, we evaluated the quantification accuracy for the 830 labeled metabolites identified by MetTracer. We manually integrated peak areas of the 7426 isotopologues from the 830 labeled metabolites using Skyline, and compared the

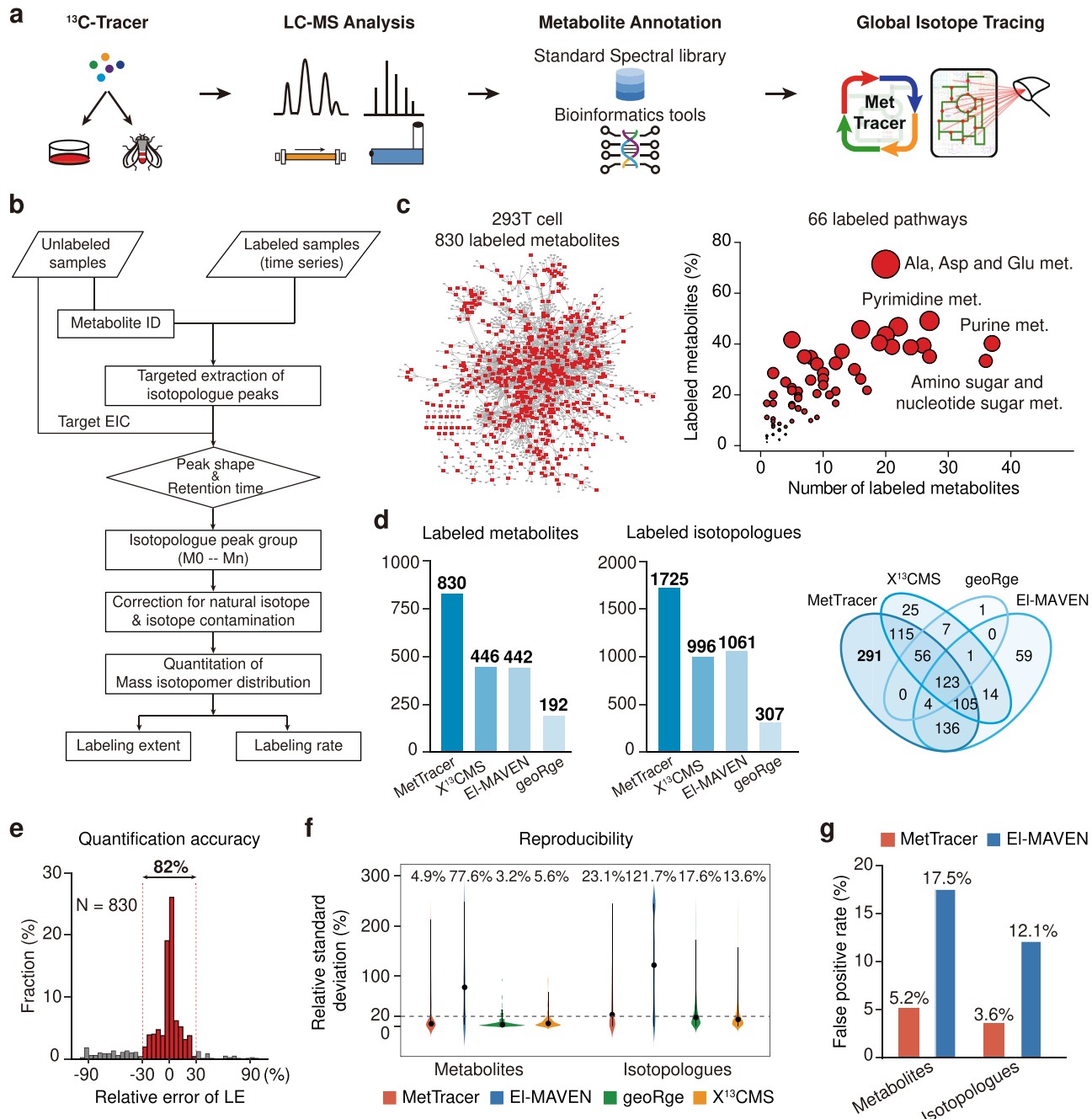

**Fig. 1 Global stable-isotope tracing metabolomics. a** Schematic illustration of MetTracer workflow. **b** Detailed data processing workflow in MetTracer. **c** Distributions of the 830 labeled metabolites in 293T cells in the metabolic network (left panel) and pathways (right panel). Red dots in the left panel represent the labeled metabolites. The circle size in the right panel represents the ratio of the number of labeled metabolites to the number of metabolites in a pathway. **d** Numbers of labeled metabolites and isotopologues using MetTracer and other indicated software tools. The Venn diagram shows the overlap of the labeled metabolites using MetTracer and other software. **e** Distribution of relative errors of labeling extent values between MetTracer and manual analysis using Skyline ($n = 830$ metabolites). **f** Relative standard deviation (RSD) distributions of metabolites and isotopologues obtained from MetTracer and other software tools ($n = 6$ technical replicates of 293T cell samples). The black dots represent median RSD. **g** False-positive rates of the labeled metabolites and isotopologues obtained from MetTracer and El-MAVEN. Source data are provided as a Source Data file.

labeling extents (LE; see Methods) with those calculated by MetTracer. Results showed that 82% of metabolites had good consistency between MetTracer and manual analysis using Skyline (relative errors ≤ 30%; Fig. 1e and Supplementary Data 5). We also demonstrated a good quantitative consistency of the labeled isotopologue abundances between MetTracer and Skyline (Supplementary Fig. 2c). Examples were given for the labeled fractions of metabolites analyzed by MetTracer and Skyline in

Supplementary Fig. 2d. To investigate the extraction reproducibility, we calculated the relative standard variations (RSDs) for the labeled metabolites and isotopologues in the stable-isotope labeled 293T cell samples. Median RSDs of labeled fractions for metabolites and isotopologues obtained using MetTracer were 4.9% and 23.1%, respectively, which were close to those from geoRge and X13CMS. As a comparison, El-MAVEN resulted in significantly higher median RSD values of 77.6% and 121.7% for

the labeled metabolites and isotopologues, respectively (Fig. 1f). Further, we evaluated the false-positive rate (FPR) of MetTracer. MetTracer outperformed El-MAVEN in FPR performance, with the FPR values of 5.2% and 3.6% for the labeled metabolites and isotopologues, respectively (Fig. 1g and Supplementary Table 2). In addtion, we have also provided several metabolite examples to demonstrate the unique technical advantages of MetTracer (Supplementary Figs. 3–6).

To demonstrate the broad application of MetTracer in different mass spectrometers, we also analyzed the stable-isotope labeled 293T cell samples using an Orbitrap mass spectrometer. Herein, a total of 1035 metabolites were annotated (347, 144, and 544 metabolites with MSI levels 1, 2, and 3, respectively), and 14,132 corresponding isotopologues were generated in 293T cell samples. With a total of 896 metabolites (86.6%) and 11,844 isotopologues (83.8%) being extracted, MetTracer identified 635 $^{13}$C-labeled metabolites and 1433 $^{13}$C-labeled isotopologues, which covered 69 metabolic pathways successfully (Supplementary Fig. 7a, d). Also, we demonstrated the technical advancements of MetTracer including the high coverage, high quantification accuracy and reproducibility, and low false-positive rate for analyzing the stable-isotope labeled 293T cell dataset from the Orbitrap mass spectrometer (Supplementary Fig. 7e–h, Supplementary Table 3, Supplementary Data 6). Altogether, MetTracer is a high-coverage isotope tracing metabolomics technology enabling global tracking of stable-isotope labeled metabolites with high reproducibility and quantification accuracy, as well as low false-positive rate.

**Quantitative in vivo stable-isotope tracing of the metabolome in *Drosophila*.** Next, we used *Drosophila* as a model organism to demonstrate the in vivo application of global isotope tracing metabolomics, and applied MetTracer to quantify the labeling extents and rates of metabolites in *Drosophila*. We conducted the continuous stable-isotope labeling in *Drosophila* using [U-$^{13}$C]-glucose and collected head tissues at six time intervals for global isotope tracing metabolomics. In head tissues, a total of 745 metabolites were annotated, and 390 labeled metabolites from 59 metabolic pathways were tracked by MetTracer (Fig. 2a and Supplementary Data 7). We also validated the high coverage, high quantification reproducibility, and low false-positive rate for labeled metabolites in *Drosophila* labeling experiment using MetTracer (Supplementary Fig. 8 and Supplementary Table 4). Then, labeling extents for each metabolite at all time points were calculated, and were subjected to hierarchical cluster analysis (HCA; Fig. 2b). Three cluster groups were generated, with 201 metabolites being categorized in cluster 1, 94 metabolites in cluster 2, and 95 metabolites in cluster 3. We next analyzed the labeled metabolites in individual clusters by the non-linear fitting of labeling extents and labeling times (Fig. 2c). The fastest labeling cluster 1 reached the isotopic steady state within 3–6 h. Pathway-enrichment analysis revealed that labeled metabolites in cluster 1 were mainly in fructose and mannose metabolism and galactose metabolism ($p$-value < 0.05; hypergeometric test). Labeled metabolites in cluster 2 exhibited slower labeling rates, which required 12–24 h to reach the isotopic steady state. These metabolites were enriched in lysine degradation. In contrast, metabolites grouped as cluster 3 wherein purine metabolism and fatty acid biosynthesis were enriched, could not reach the isotopic steady state after the 24-h labeling.

We then fitted the in vivo tracing kinetics of labeled metabolites in *Drosophila* using the general first-order exponential function, and enabled large-scale quantitation of labeling rates from the [U-$^{13}$C]-glucose tracer to the downstream metabolites (see Methods). Labeling extents for each metabolite at all time points were used to fit the exponential function. Thus, the labeling rate $k$, which is the apparent first-order constant in the exponential function, was calculated to quantify the incorporation rate of the tracer to metabolite targets in the continuous stable-isotope labeling. Taking aspartate as an example, the labeling extent was gradually increased to 0.84 within 24 h labeling. The fitted labeling rate $k$ of aspartate was calculated as 0.12 h$^{-1}$ with an $R$-value of 0.99 (Fig. 2d). In total, labeling rates for 292 out of 390 labeled metabolites were successfully fitted with $R$-values > 0.8 (Supplementary Data 8). Comparative analysis showed that $k$-values were significantly different among three cluster groups, with cluster 1 being the largest $k$-value (0.67 h$^{-1}$) and cluster 3 being the smallest $k$-value (0.08 h$^{-1}$) calculated (Fig. 2e). Other metabolite examples in each cluster with labeling rates $k$-values were also provided (Supplementary Fig. 9). We further investigated the mean labeling rates for metabolites from 19 metabolic pathways in *Drosophila* head, and provided an estimation of metabolic kinetics on the pathway level. (Fig. 2f). Generally, the mean labeling rates $k$-values were decreased from carbohydrate metabolism, to amino acid metabolism, and down to purine and fatty acid metabolism. Galactose metabolism (dme00052) was found as the pathway with the highest labeling rate, and fatty acid biosynthesis (dme00061) was the lowest one. Collectively, we demonstrated that the application of MetTracer in the continuous stable-isotope labeling of *Drosophila* provided in vivo and quantitative characterizations of labeling kinetics from the tracer with a metabolome-wide coverage.

**Quantitative stable-isotope tracing reveals distinct metabolic profiles in different tissues.** We next sought to compare the inter-tissue differences in metabolic activity between head and muscle tissues in *Drosophila* using global isotope tracing metabolomics. In the same continuous labeling experiment, MetTracer tracked 597 labeled metabolites from 64 metabolic pathways in *Drosophila* muscle tissue (Supplementary Fig. 10a, Supplementary Data 7). HCA analysis also identified three metabolite clusters in the muscle (Fig. 3a). Non-linear fitting analysis of labeled metabolites in each cluster revealed three different labeling profiles, and different metabolic pathways were enriched in clusters 1–3 (Fig. 3b). Similarly, labeling rates $k$-values were calculated and clear differences among three clusters were observed (Supplementary Fig. 10b and Supplementary Data 8). With labeling extent being quantitatively characterized, we systematically compared the metabolic labeling extents of 14 shared metabolic pathways between head and muscle tissues (Fig. 3c and Supplementary Fig. 11a–c). Interestingly, energy metabolism related pathways such as pentose phosphate pathway displayed higher labeling extents and activities in the muscle than in the head. As a comparion, labeling extents of metabolites in pathways such as nicotinate and nicotinamide metabolism, amino acid metabolisms, including alanine, histidine, cysteine and methionine metabolism, and purine metabolism were presented at higher levels in the head compared to the muscle (Fig. 3c). Taken the purine metabolism as an example (Fig. 3d, e and Supplementary Fig. 11d), intermediate metabolites including adenosine monophosphate (AMP), adenosine, adenine, inosine, and guanosine displayed higher labeling extents and labeling rates in the head than those in the muscle. The higher metabolic activity of purine metabolism in the head was also evidenced by the significantly higher labeled fractions of isotopologues ($M + 5$, $M + 6$, $M + 7$, $M + 8$, $M + 9$, and $M + 10$; Fig. 3e). Altogether, in vivo isotope tracing metabolomics supported quantitative comparison of metabolic activities across tissues and revealed distinct inter-tissue metabolism in *Drosophila*.

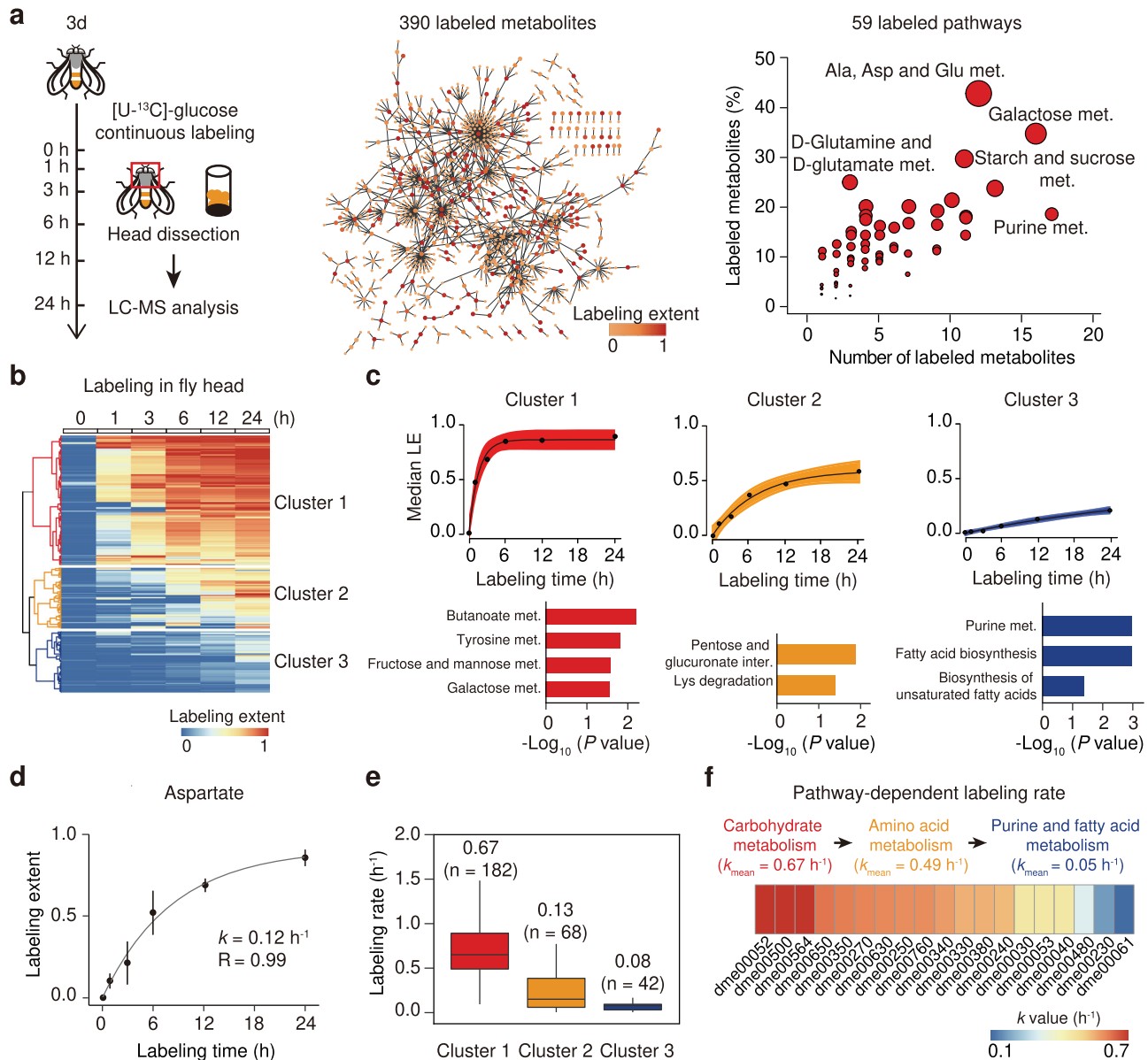

**Fig. 2 In vivo stable-isotope tracing of the metabolome in *Drosophila*. a** Illustration of in vivo stable-isotope tracing in *Drosophila* and the distribution of 390 labeled metabolites from 59 pathways in head tissues. **b** Hierarchical-clustering analysis of the labeled metabolites ($n = 390$). **c** Upper panels, labeling dynamics of metabolites in each cluster; black dots represent the median labeling extent (LE) values of metabolites in the cluster; black lines represent the fitted metabolic dynamics; error bands represent 95% confidence intervals. Bottom panels, enriched pathways in each cluster ($p$-values < 0.05; hypergeometric test). **d** The example of aspartate for the calculation of labeling rate $k$ ($n = 10$ biological replicates in each time point); black dot represents median LE; whiskers represent ±s.d. **e** $k$-value distributions for metabolites in three clusters ($n = 182, 68, 42$ fitted metabolites in clusters 1, 2, and 3, respectively). The centerlines of the boxplots indicate the median values, the lower and upper lines in boxplots correspond to 25th and 75th quartiles, and the whiskers indicate the largest and lowest points inside the range defined by 1st and 3rd quartile plus 1.5 times interquartile ranges (IQR). **f** Heat map showing the mean $k$-values of metabolties from 19 significantly labeled pathways. Pathway names are provided in Supplementary Table 5. Source data are provided as a Source Data file.

**System-wide loss of metabolic coordination in *Drosophila* during aging**. Dysregulation in metabolic homeostasis represents a hallmark of aging. Metabolic coordination between metabolites is crucial for the maintenance of tissue metabolic homeostasis, particularly in the context of aging. Here, we characterized the system-wide alternations in metabolic coordination in both head and muscle tissues during *Drosophila* aging (Fig. 4a). In head tissues, we performed Pearson correlation analyses of labeling extents among the 390 labeled metabolites (Fig. 4b). We observed as many as 1117 significant metabolite–metabolite correlations in young head tissues (3d).

Strikingly, these correlations were dramatically reduced by 38% in old head tissues (30d). Taken the labeled glucose and pyruvate as examples, the labeling extents of glucose were positively correlated with those of pyruvate in young head tissues, whereas the correlation was completely lost in old ones (Fig. 4c and Supplementary Fig. 12a). This loss of metabolic coordination is consistent with our previous report that a progressive decline of glycolysis is associated with aging in *Drosophila*[41]. In addition, the transcriptional levels of the glycolytic genes were also decreased in the old head tissues, which further explained the loss of metabolic coordination between glucose and

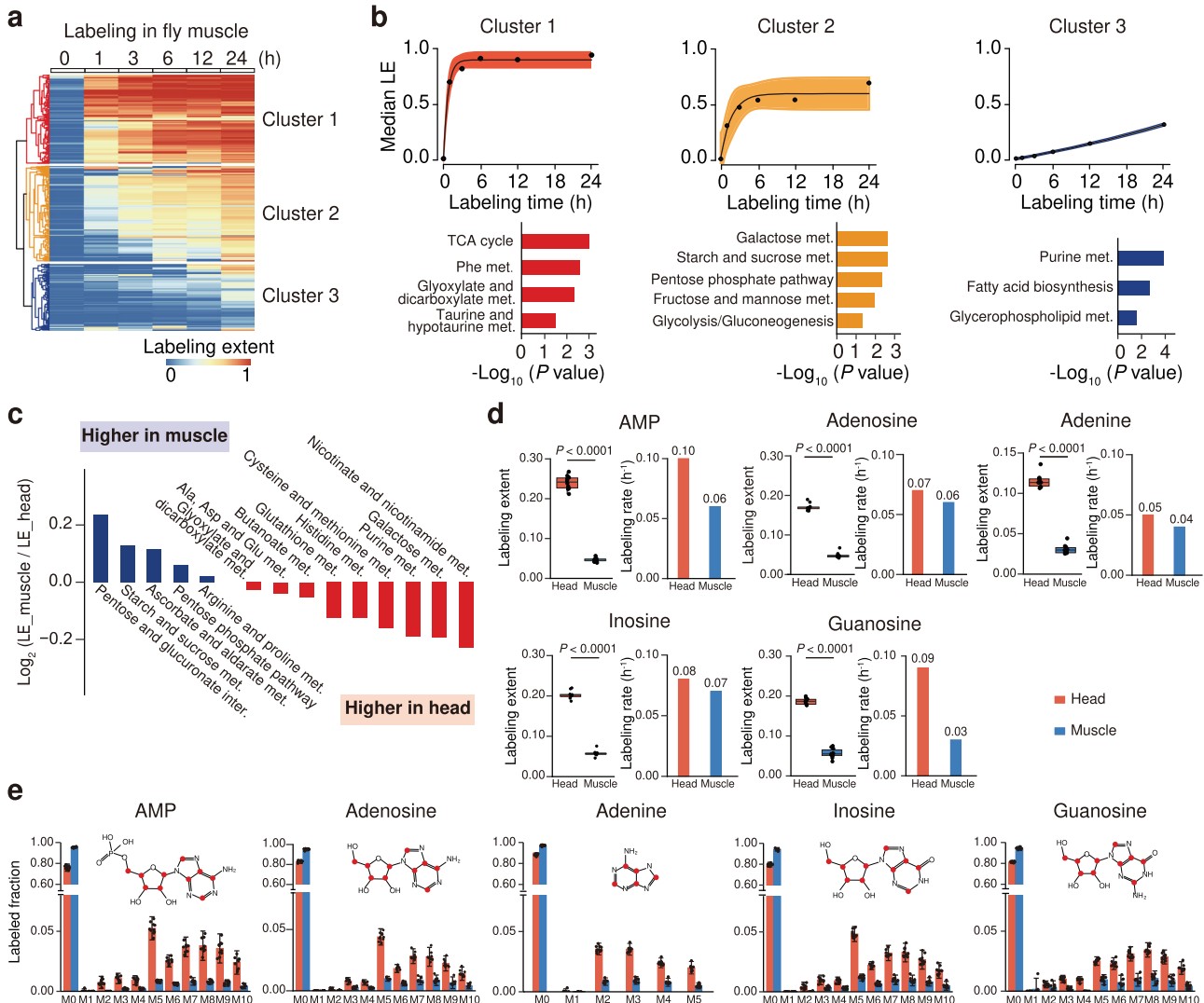

**Fig. 3 Quantitative stable-isotope tracing reveals distinct metabolic profiles in different tissues. a** Hierarchical-clustering analysis of labeled metabolites in *Drosophila* muscle tissues ($n = 597$ metabolites). **b** Upper panels, labeling dynamics of metabolites in each cluster; black dots represent the median labeling extent (LE) values of metabolites in the cluster in muscle tissue; black lines represent the fitted metabolic dynamics; error bands represent 95% confidence intervals. Bottom panels, enriched pathways in each cluster ($p$-values < 0.05; hypergeometric test). **c** Comparisons of labeling extents of 14 shared metabolic pathways in head and muscle tissues. **d** Labeling extents (LE) and labeling rates ($k$) of metabolites in purine metabolism ($n = 10$ biological replicates per group; two-tailed Student's $t$-test). The centerlines of the boxplots indicate the median values; the lower and upper lines in boxplots correspond to 25th and 75th quartiles, and the whiskers indicate the largest and lowest points inside the range defined by 1st and 3rd quartile plus 1.5 times interquartile ranges (IQR). **e** Labeled fractions of metabolite isotopologues in purine metabolism. Bar plots represent mean ± SD ($n = 10$ biological replicates per group). The red dots in chemical structures represent ^13C-labeled carbon atoms. Source data are provided as a Source Data file.

pyruvate (Supplementary Fig. 12b). We further classified the metabolite–metabolite correlations according to their labeling rate clusters shown in Fig. 2 (Fig. 4b). Examination of metabolic coordination in individual metabolite cluster revealed that metabolites in the fast-labeling cluster 1 were severely affected by aging, wherein 42% of correlations were diminished (Fig. 4b, right panel). Similarly, substantial loss of metabolite correlations was observed in the *Drosophila* muscle during aging, with the total correlations being decreased from 1,565 to 822 (47%; Fig. 4d). Again, metabolites in the fast-labeling cluster 1 were severely affected by aging, wherein as large as 78% of correlations were diminished in old muscle tissues. We also analyzed correlations between the metabolites in the head and muscle tissues. The inter-tissue metabolic coordination was also impacted by aging, with the number of correlations being significantly decreased (Fig. 4e). Metabolites in cluster 1 lost the

largest number of intra-tissue correlations, with the total correlations being decreased by 65%. We also investigated the effect of aging on inter-cluster metabolite correlations. In fly head and muscle tissues, the correlations between metabolites in fast-labeling cluster 1 and other clusters were severely affected by aging, wherein about 60% correlations were diminished. The correlations between metabolites in cluster 2 and cluster 3 were slightly affected by aging within head and muscle tissues (Supplementary Fig. 13a, b). The inter-cluster metabolic coordination between head and muscle was also impacted by aging, especially metabolites in cluster 3 with other clusters, wherein >60% of correlations were diminished (Supplementary Fig. 13c). In summary, in vivo isotope tracing metabolomics revealed a system-wide loss of metabolic coordination that significantly impacted both intra- and inter-tissue metabolic homeostasis in aging *Drosophila*.

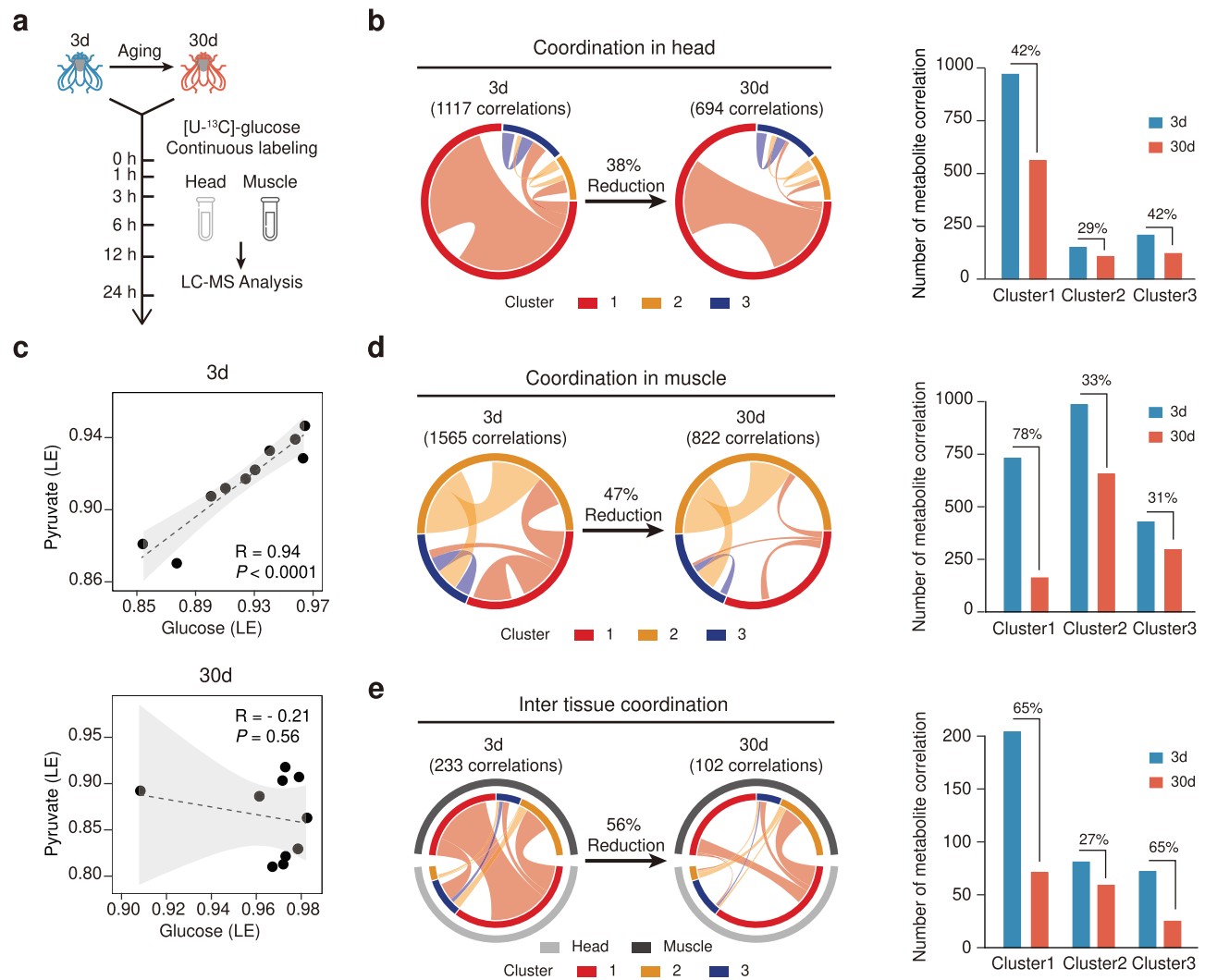

**Fig. 4 System-wide alternations of metabolic homeostasis in *Drosophila* during aging. a** The in vivo isotope tracing metabolomics of young (3d) and old (30d) *Drosophila*. **b** Left, a Circos plot showing metabolite–metabolite correlations in young and old *Drosophila* head tissues. Right, a bar plot showing the numbers of metabolite–metabolite correlations in each metabolite cluster. **c** Correlation of labeling extents for the labeled glucose and pyruvate during aging ($n = 10$ biological replitates in each group). Dashed line, linear regression; gray shadow, 95% confidence interval of the linear relationship between pyruvate and glucose. The *p*-value of Pearson correlation coefficient was calculated by two-sided Student's *t*-test. **d** Left, a Circos plot showing metabolite–metabolite correlations in young and old *Drosophila* muscle tissues. Right, a bar plot showing the numbers of metabolite–metabolite correlations in each metabolite cluster. **e** Left, a Circos plot showing metabolite–metabolite correlations between head and muscle tissues in young and old *Drosophila*. Right, a bar plot showing the numbers of metabolite–metabolite correlations in each metabolite cluster. In panels **b**, **d**, **e**, the colored circle refers to the labeling rate cluster. Clusters 1–3 represent the labeling rate clusters in Figs. 2 and 3. Ribbon connecting metabolite clusters refers to the significant correlations between metabolites. Ribbon thickness refers to the number of significant metabolite-to-metabolite correlations. Source data are provided as a Source Data file.

**Metabolic shift from glycolysis to serine metabolism in aging *Drosophila*.** Pyruvate, the end product of aerobic glycolysis, has been demonstrated as a crucial metabolic hub for multiple pathways to sustain metabolic homeostasis in organisms. In young *Drosophila* head tissues, after the 24-h labeling, pyruvate had metabolic correlations with other 23 metabolites such as serine, succinate, and glucose (Supplementary Table 6). Interestingly, these metabolic correlations with pyruvate were completely disrupted in old *Drosophila* (Fig. 5a). In particular, in young *Drosophila* but not old ones, pyruvate was positively correlated with serine, which is an important carbon donor to the one-carbon metabolism. Since serine can be metabolically converted to glycine by serine hydroxymethyl transferase (*shmt*), we next examined the relationship between pyruvate and glycine in the two age groups. Results showed that pyruvate was also

positively correlated with glycine in young *Drosophila* head, while this correlative relationship was not present in the aged ones (Fig. 5b). Notably, the correlation scenario between pyruvate and serine and glycine during aging could not be revealed by abundance correlation analysis in unlabeled *Drosophila* samples (Supplementary Fig. 14a).

We also analyzed the labeling extents of pyruvate, serine and glycine in the two *Drosophila* groups. Pyruvate showed decreased labeling extent in aged *Drosophila* while serine and glycine had higher labeling extents (Fig. 5c). Comparisons of $^{13}$C-labeled fractions revealed that the labeled fraction of pyruvate ($M + 3$) was decreased after 24-h isotope labeling in the old *Drosophila*, while that of serine ($M + 3$) was increased. In 30d *Drosophila*, glycine was isotopically labeled rapidly after 1 h feeding of [U-$^{13}$C]-glucose, and its labeled fraction ($M + 2$) was significantly

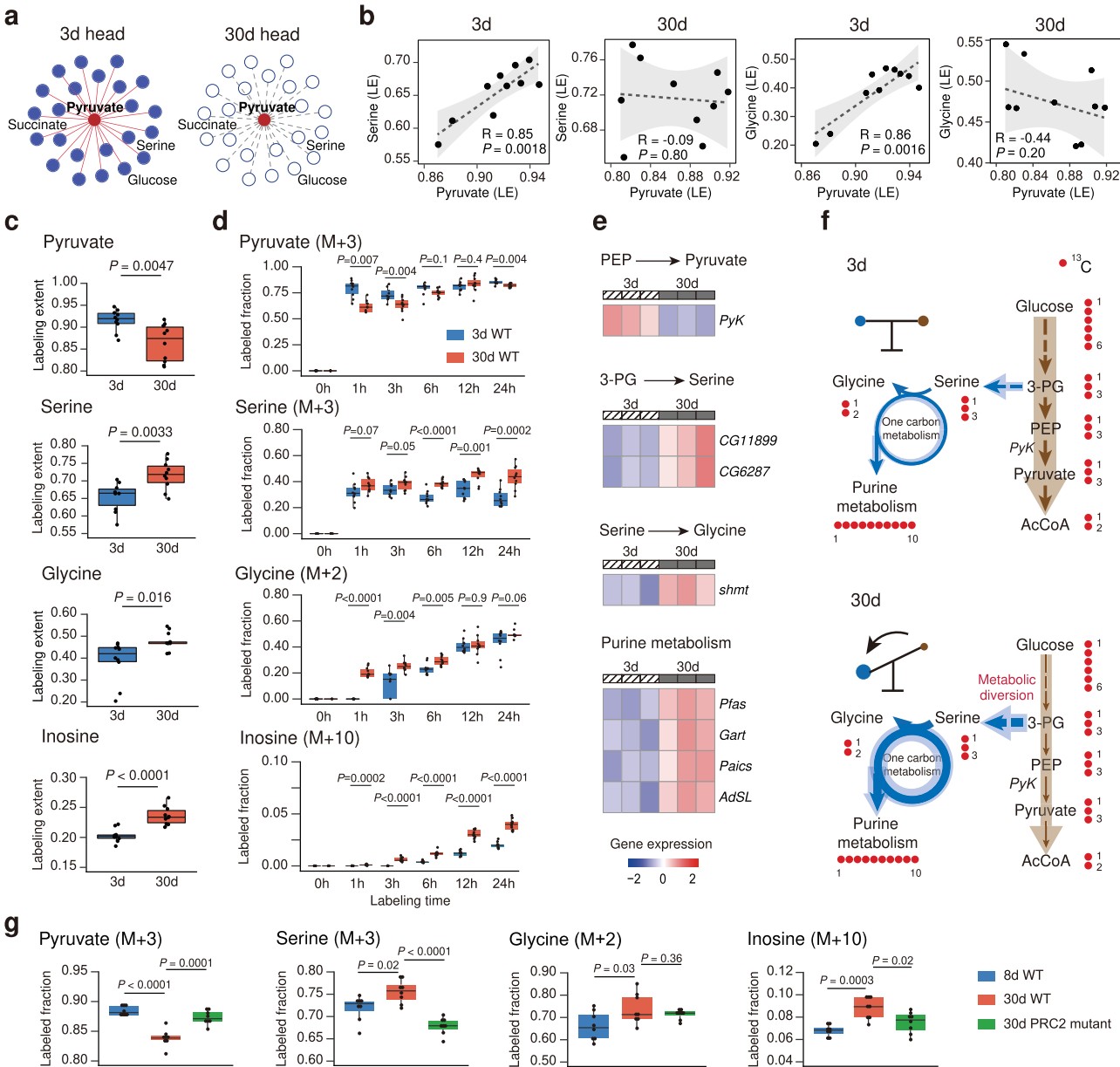

**Fig. 5 Metabolic shift from glycolysis to serine metabolism in aging *Drosophila*. a** Correlation network showing the correlation between labeled pyruvate and other labeled metabolites in *Drosophila* head tissue. Red line, significant correlation; gray dashed line, no correlation. **b** Correlations between labeled pyruvate and serine or glycine during aging ($n = 10$ biological replicates in each group). Dashed line, linear regression; gray shadow, 95% confidence interval of the linear relationship between pyruvate and serine or glycine. The *p*-value of Pearson correlation coefficient was calculated by two-sided Student's *t*-test. **c** Labeling extents (LE) of metabolites in 3d and 30d *Drosophila* ($n = 10$ biological replicates per group; two-tailed Student's *t*-test). **d** Boxplots showing the labeled fractions of pyruvate ($M + 3$), serine ($M + 3$), glycine ($M + 2$) and Inosine ($M + 10$) at each time point in 3d and 30d WT flies ($n = 10$ biological replicates in each group). **e** Heat maps of the *z*-score normalized gene expression levels of enzymes. The significantly changed genes were shown (*P*-value < 0.05; two-tailed Student's *t*-test). **f** The proposed model of metabolic rewiring in aging *Drosophila*. **g** Boxplots showing the labeled fractions of pyruvate ($M + 3$), serine ($M + 3$), glycine ($M + 2$), and Inosine ($M + 10$) in 8d WT, 30d WT, and 30d *Drosophila* with PRC2 mutation ($n = 8$ biological replicates in each group). In **d, g**, the labeled fraction was normalized to the fraction of glucose ($M + 6$). In **c, d, g**, the centerlines of the boxplots indicate the median values; the lower, and upper lines in boxplots correspond to 25th and 75th quartiles, and the whiskers indicate the largest and lowest points inside the range defined by 1st and 3rd quartile plus 1.5 times interquartile ranges (IQR). Two-tailed Student's *t*-test. Source data are provided as a Source Data file.

higher than that in 3d *Drosophila* (Fig. 5d). One-carbon metabolism fueled by serine biosynthesis further contributes to purine metabolism. We found that the representative metabolites in purine metabolism showed higher labeling extents (Fig. 5c and Supplementary Fig. 14b) and labeled fractions of metabolites such as inosine ($M + 10$), inosine monophosphate (IMP, $M + 10$), adenosine monophosphate (AMP, $M + 10$), adenosine ($M + 10$),

and adenine ($M + 5$) in old *Drosophila* head tissues than those in young ones (Fig. 5d and Supplementary Fig. 14c), which suggests a higher metabolic activity of purine metabolism in old *Drosophila*. Importantly, these results revealed by the global isotope tracing metabolomics were evidenced by the levels of genes responsible for these pathways (Fig. 5e and Supplementary Data 9). *Pyk*, which is the gene for pyruvate kinase in glycolysis to

produce pyruvate from phosphoenolpyruvate (PEP), was significantly reduced in 30d *Drosophila*. Glucose metabolism provides the precursor 3-phosphoglycerate (3-PG), thereby supporting serine biosynthesis. We found that two genes involved in serine biosynthesis, *CG11899* and *CG6287*, were significantly increased in 30d *Drosophila*. Higher levels of *shmt* in old *Drosophila* heads were also observed. In addition, genes responsible for purine metabolism, including *Pfas*, *Gart*, *Paics*, and *AdSL* showed increased levels in 30d *Drosophila*. Altogether, these data revealed that declined glycolysis activity during aging was coupled with enhanced purine metabolism that was fueled by serine and glycine supported one-carbon metabolism. Finally, we propose a metabolic rewiring model during *Drosophila* aging, wherein glucose is metabolically channeled to one-carbon metabolism and purine metabolism from glycolysis (Fig. 5f).

To verify the metabolic remodeling, we investigate whether the metabolic shift from glycolysis to serine metabolism and purine metabolism in aged *Drosophila* could be reverted in the long-lived *Drosophila* mutation. In our previous report, the PRC2 mutation has been proven to extend the lifespan of *Drosophila* by boosting glycolysis in aged *Drosophila*[41]. We re-analyzed the data from the [U-$^{13}$C]-glucose labeling experiments in young (8d) and aged (30d) wild-type (WT) *Drosophila*, and aged (30d) PRC2 mutants for 24 h. The results demonstrated that the labeled fraction of pyruvate ($M + 3$) was decreased with aging in wild-type *Drosophila* but rescued in the age-matched PRC2 mutants (Fig. 5g). In addition, the increased labeled fractions of serine ($M + 3$) and inosine ($M + 10$) were also significantly reversed by PRC2 mutation. The results suggested that reversing the metabolic shift from glycolysis to one-carbon metabolism and purine metabolism may contribute to the lifespan extension in *Drosophila*.

## Discussion

Untargeted metabolomics measures metabolite concentrations on a large scale and provides system-wide characterizations of metabolic status in living systems. However, changes in metabolite concentrations do not readily imply metabolic activities. Isotope tracing technology enables unraveling metabolic activity, but is largely restricted by the metabolite coverage. In this work, we developed MetTracer, an untargeted isotope tracing metabolomics technology, to trace labeled metabolites with a metabolome-wide coverage. We benchmarked that MetTracer has made pivotal improvements in terms of coverage, quantification accuracy, reproducibility and false-positive rate. Although we have tried our best to perform fair comparisons between different software tools, and provided the detailed parameters for each tool, such comparisons are still difficult to judge, because output of each tool being compared may depend significantly on the choice of tool-specific parameters. The main advancement of MetTracer comes from the integration of untargeted metabolite annotation and targeted isotopologue extraction. First of all, MetTracer benefited from the input of metabolite annotations of unlabeled samples from different bioinformatic tools, which ensures the high coverage of labeled metabolite extraction. Second, MetTracer has also made several data processing innovations to targeted isotopologue extraction, which ultimately contributes to the high coverage of labeled metabolites and high accuracy of isotopologue quantitation. For example, MetTracer performed targeted extraction of all possible isotopologues and employed the most intensive isotopologue peak instead of the M0 base peak for peak detection. For highly labeled metabolites wherein M0 peak has a relatively low intensity, the use of the most intensive isotopologue peak achieved a higer rate of successful peak detection. Then, the employment of targeted extracted ion chromatogram (i.e., target

EIC in Fig. 1b; see Methods) from unlabeled samples and calculation of peak-peak correlation (PPC) between isotopologue peak and target EIC peak also improved the accuracy and reduced the false positives. Finally, the isotope contamination estimation and correction improved accurate quantitation of isotopologues. We have provided several examples to demonstrate the unique technical advantages of MetTracer (Supplementary Figs. 3–6). It is also worthy to note that MetTracer supports data analysis of multiple sample groups simultaneously. Therefore, MetTracer could be of enormous application values in a broad range of biological scenarios wherein more than two groups are generally compared.

MetTracer enabled a system-wide measurement of several hundreds of isotopically labeled metabolites and harvested rich information on labeling patterns, labeling extents, and labeling rates in one experiment, thereby enabling characterization of metabolic activities at systems-level. Conventionally, stable-isotope tracing technology resolves quantitative metabolic fluxes using metabolic flux analysis (MFA) and flux balance analysis (FBA). However, it requires comprehensive prior knowledge of biochemical reactions in specific pathways, and is very time and computationally intensive. More challenging is to calculate fluxes for hundreds of metabolites in a pathways-intertwined metabolic network. These challenges are particularly prominent in mammalian systems for fluxes analyses, and most of existed quantitative flux studies were limited to specific pathways and simple model organisms (such as yeast and bacteria). Previously, Yuan et al. developed the kinetic flux profiling (KFP) analysis for metabolic flux calculation, which required a fast switch of labeling substrates[42]. As such, the KFP model is typically used in cultured cells wherein the labeling substrate in media can be rapidly exchanged. Here, the KFP model for metabolic flux calculation may not be valid for in vivo isotope tracing of *Drosophila*. Instead, we fitted the in vivo tracing kinetics of labeled metabolite using the first-order exponential equation to calculate labeling rate $k$-value. The labeling rate k quantifies the incorporation rate of the [U-$^{13}$C]-glucose tracer to downstream metabolite targets in the continuous stable-isotope labeling. Our data showed that, in the head tissue, 75% metabolites (292 out of 390) had the in vivo tracing kinetics successfully fitted with the first-order exponential equation. Similarly, 73% metabolites (437 out of 597) in the muscle tissue had successful fittings. Since the $k$-value indicates the incorporation rate of the tracer to a metabolite target, it is reasonable that the isotopically labeled metabolites nearer to the [U-$^{13}$C]-glucose tracer in the metabolic network have higher $k$-values being calculated. Indeed, we showed that metabolites in glycolysis such as pyruvate exhibited larger $k$-values than those in fatty acid synthesis such as hexadecenoic acid (Supplementary Fig. 15). Of note, we cannot determine the absolute metabolic flux through multiplying the $k$-values calculated here by metabolite concentrations because the assumption of KFP analysis is not met in *Drosophila*. This limitation, however, does not detract from our fittings of in vivo tracing kinetics on a metabolome-wide scale and comparative analyses of $k$-values for the same metabolite between different biological scenarios.

*Drosophila* is a widely used model organism to study aging. Previous studies using traditional metabolomics methods have disclosed metabolic changes associated with aging in varied genotypes of *Drosophila*[35,40,41,43–53] (Supplementary Table 7). For example, Chang et al.[54], Hoffman et al.[45], and our group[41] reported the declination of metabolites in glycolysis pathway during aging. Metabolites in amino acid metabolism were found to have correlation with aging[46–49,52,53]. For example, Tapia et al. revealed that reduction in amino acids such as histidine, tyrosine, and leucine is protective against aging[52]. Instead, Parkhitko et. al. showed that metabolic intervention with supplementation of

tyrosine contributed to lifespan extension[48]. In addition, Laye et al. reported that S-adenosyl-methionine (SAM) cycle was downregulated in flies under diet restricted condition compared to the age-matched flies with normal diet, suggesting that the long-lived flies have lower level of SAM[46]. Similarly, enhancement of SAM catabolism by overexpressing glycine N-methyltransferase (*Gnmt*) has also been demonstrated to extend lifespan of *Drosophila*, as reported by Obata et al.[35] Elevated purine metabolism was also found to be associated with short lifespan of *Drosophila*[40,50].

Despite the accumulative aging-related metabolites and metabolic pathways being uncovered by measuring the changes of metabolite concentrations associated with aging, these were independently studied, and the underlying mechanism by which coordination of metabolic activities impacts aging is not elucidated due to the lack of appropriate technologies. Here, in addition to metabolite concentrations, we demonstrated that the global isotope tracing metabolomics can provide an orthogonal perspective to the changed metabolome by monitoring the labeling parameters at different levels, thereby supporting quantitative comparison of metabolic activities during aging. We discerned the age-dependent changes of metabolic activities in *Drosophila*, in which carbohydrate related metabolic pathways were enriched with metabolites of declined metabolic activities in aging, while amino acid related metabolic pathways and purine metabolism were enriched with metabolites of increased metabolic activities in aging, which were consistent with the observations in previous studies (Supplementary Table 7). This suggests that distinct metabolite utilization in different biological scenarios can explain the age-dependent metabolic responses in physiological status. A key contribution of this work is to reveal a system-wide loss of metabolic coordination during aging in *Drosophila* using global isotope tracing metabolomics. This uniquely system-wide loss was evidenced by the massive reduction of metabolite–metabolite correlations within the head and muscle tissues and across tissues. We showed that glycolysis was skewed to one-carbon metabolism that ultimately fueled purine metabolism, suggesting that the disruption of metabolic coordination among three metabolic pathways promotes *Drosophila* aging. In agreement with our previous study, with the long-lived *Drosophila* (PRC2 mutants) that have enhanced glycolysis activity[41], we revealed that the PRC2 mutation had decreased metabolic activities for intermediate metabolites in one-carbon metabolism and purine metabolism compared with the age-matched wild-type *Drosophila*. This result, combined with our findings of the system-wide loss of metabolic coordination observed in naturally aging *Drosophila*, highlights the importance of metabolic homeostasis and coordination among glycolysis, one-carbon metabolism, and purine metabolism as a critical regulation of aging and longevity.

In this study, we used *w1118* fly model to demonstrate the utility of MetTracer in studying aging, and we considered that, the metabolic changes between 3d flies and 30d flies were caused by aging here. Although the *w1118* flies were widely used as wild-type controls in many fly aging studies[41,46,47,51], it is worth noting that, previous studies have discovered that the *w* mutant of *w1118* flies have an impact on fly metabolism. The *Drosophila w* gene encodes an ATP binding cassette transporter, which contributes to transportation of metabolites such as guanine, tryptophan and kynurenine[55–58]. The *w* mutant flies were also observed with low levels of the biogenic amines, such as serotonin, dopamine, and histamine[59,60]. The *Drosophila w* gene is expressed principally in eyes and excretory organs and testes, but has very low levels observed in the glia and neurons of the brain and various other tissues[61,62]. Although whether the metabolome of *w1118* will be affected by aging is unclear, we should still be cautious that this mutation alters the metabolism of *Drosophila*.

## Methods

**Chemicals.** LC–MS grade water (H₂O) was purchased from Honeywell (Muskegon, MI, USA). LC–MS grade acetonitrile (ACN) was purchased from Merck (Darmstadt, Germany). Ammonium hydroxide (NH₄OH) and ammonium acetate (NH4OAc) were purchased from Sigma (St. Louis, MO, USA). The [U-$^{13}$C]-glucose, [U-$^{13}$C]-glutamine and [U-$^{13}$C]-acetate was purchased from Cambridge Isotopes laboratories (MA, USA).

**Stable-isotope tracing experiments.** The 293T cell line was bought from ATCC with Product No. CRL-2925. 293T cells were seeded in 6-cm cell culture plates with Dulbecco Modified Eagle's Medium (DMEM) containing with 10% dialyzed fetal bovine serum (dFBS) and 1% penicillin/streptomycin. When cells were grown to 80% confluence, unlabeled DMEM was removed. Cells were washed using ~3 mL of PBS. The culture medium was replaced as glucose-free and glutamine-free DMEM containing 25 mM [U-$^{13}$C]-glucose, 4 mM [U-$^{13}$C]-glutamine, 5 mM [U-$^{13}$C]-acetate and 10% dialyzed FBS. After 17 h labeling, cells were washed twice with PBS. Cell plates were placed on the dry ice and fast quenched with 1 mL of MeOH:ACN:H₂O (2:2:1, v/v/v, pre-cooled in −80 °C refrigerator) solvent mixture. The plates were incubated at −80 °C for 40 min. Cells were scraped from the plate and transferred to a 1.5-mL centrifuge tube. The samples were vortexed for 1 min, and followed by 15 min centrifugation using 16,200 × g at 4 °C. The supernatant was taken and evaporated to dryness in a vacuum concentrator. Dry extracts were reconstituted in 100 µL of ACN:H₂O (1:1, v/v), followed by 10 min sonication (50 Hz, 4 °C) and 15 min centrifugation using 16, 200 × g at 4 °C to remove insoluble material. Supernatants were finally transferred to HPLC glass vials and stored at −20 °C prior to LC/MS analysis.

The fly culture and stable-isotope labeling experiments followed our previous publications[63]. In brief, flies were cultured in standard media at 25 °C with 60% humidity in a 12 h light and 12 h dark cycle. The standard *Drosophila* food contains sucrose (36 g/L), maltose (38 g/L), yeast (22.5 g/L), agar (5.4 g/L), maizena (60 g/L), soybean flour (8.25 g/L), sodium benzoate (0.9 g/L), methyl-p-hydroxybenzote (0.225 g/L), propionic acid (6.18 mL/L), and H₂O to make up 1 L of the food. The wild-type fly line used was 5905 (FlyBase ID FBst0005905, w1118). To age flies, adult male flies were collected at the day of eclosion and maintained at 20 flies per vial at 25 °C with 60% humidity and a 12 h light and 12 h dark cycle. Aged flies were transferred to new vials every other day. For isotope tracing study, male flies at ages of 3d and 30d were used with 15 flies per vial. Prior to isotope labeling, flies were starved for 6 h, then transferred to new vials containing a small piece of Kimwipe paper pre-soaked in 800 µL of 10% [U-$^{13}$C]-glucose. When the desired time points (0, 1, 3, 6, 12, and 24 h) were reached, the flies were dissected under CO₂ anesthesia. It took about 15–20 s to dissect a fly. The whole muscle tissues and whole head tissues were collected from flies. Head or muscle tissues from fifteen flies were pooled into one microcentrifuge tube as one biological replicate. After dissection, the tissue samples were quickly frozen using liquid nitrogen, and stored at −80 °C. The pooled sample was used as one biologically independent replicate. For each condition, 10 biologically independent replicates were prepared. The sample heterogeneity was confirmed similar between 3d and 30d flies (Supplementary Fig. 16). The raw data of validation experiment with PRC2 mutants was retrieved from our previous study[41]. PRC2 mutation is referred to mutations to genes in the Polycomb repressive complex 2. The genotype of PRC2 mutant was *Pcl* c421/+; *Su(z)12* c253/+. Three groups of flies (WT 8d, WT 30d, PRC2 mutant 30d) were included in data re-analysis and there were eight biologically independent replicates in each group.

For metabolite extraction, dissected fly tissues were homogenized with 200 µL of H₂O and 20 ceramic beads (0.1 mm) using the homogenizer. Samples were extracted with 800 µL of ACN:MeOH (1:1, v/v), and followed by 10 min sonication (50 Hz, 4 °C). To precipitate protein, samples were incubated for 2 h at −20 °C, followed by 15 min centrifugation using 16,200 × g at 4 °C. The supernatant was taken and evaporated to dryness in a vacuum concentrator. Dry extracts were reconstituted in 100 µL of ACN:H₂O (1:1, v/v), followed by 10 min sonication (50 Hz, 4 °C), and 15 min centrifugation using 16,200 × g at 4 °C to remove insoluble material. Supernatants were transferred to HPLC glass vials and stored at −20˚C prior to LC/MS analysis.

**LC-MS analysis.** Metabolomics data of 293T cell samples were acquired using a UHPLC system (UltiMate 3000, Thermo Scientific) coupled to an orbitrap mass spectrometer (Exploris 480, Thermo Scientific). Waters ACQUITY UPLC BEH Amide column (particle size, 1.7 µm; 100 mm (length) × 2.1 mm (i.d.)) and Phenomenex Kinetex C18 column (particle size, 2.6 µm; 100 mm (length) × 2.1 mm (i.d.)) were used for LC separation for HILIC mode and RP mode, respectively. Column temperature were both kept at 25 °C. For HILIC mode, Mobile phases A = 25 mM ammonium acetate and 25 mM ammonium hydroxide in 100% water, and B = 100% acetonitrile, were used for both ESI positive and negative modes. The linear gradient eluted from 95% B (0.0–0.5 min), 95% B to 65% B (0.5–7.0 min), 65% B to 40% B (7.0–8.0 min), 40% B (8.0–9.0 min), 40% B to 95% B (9.0–9.1 min), then stayed at 95% B for 2.9 min. The flow rate was 0.5 mL/min. The sample injection volume was 2 µL. For RP mode, Mobile phases A = 0.01% acetic acid in 100% water, and B = acetonitrile/isopropanol (1/1; v/v), were used for both ESI positive and negative ionization modes. The linear gradient eluted from 1% B (0.0–1.0 min), 1% B to 99% B (1.0–8.0 min), 99% B (8.0–9.0 min), 99% B to 1% B

(9.0–9.1 min), then stayed at 95% B for 2.9 min. The flow rate was 0.3 mL/min. The sample injection volume was 2 μL. ESI source parameters were set as follows: spray voltage, 3500 V or −2800 V, in positive or negative modes, respectively; aux gas heater temperature, 350 °C; sheath gas, 50 arb; aux gas, 15 arb; capillary temperature, 400 °C. LC–MS data acquisition was operated under full scan polarity switching mode for all samples. A ddMS2 scan was appied for QC samples to acquire MS/MS spectra. The full scan was set as: orbitrap resolution, 60,000; AGC target, 1e6; maximum injection time, 100 ms; scan range, 70–1200 Da. The ddMS2 scan was set as: orbitrap resolution, 30,000; AGC target, 1e5; maximum injection time, 60 ms; scan range, 50–1200 Da; top N setting, 6; isolation width, 1.0 $m/z$; collision energy mode, stepped; collision energy type, normalized; HCD collision energies (%), 20,30,40; Dynamic exclusion duration was set as 4 s for excluding after 1 time.

Metabolomics data of 293T cell samples and fly tissue samples were also acquired using a UHPLC system (1290 series, Agilent Technologies) coupled to a quadruple time-of-flight mass spectrometer (TripleTOF 6600, Sciex). Waters ACQUITY UPLC BEH Amide column (particle size, 1.7 μm; 100 mm (length) × 2.1 mm (i.d.)) was used for the LC separation and LC condition was the same as that described above. ESI source parameters were set as followings: ion source gas 1 (GS1), 60 psi; ion source gas 2 (GS2), 60 psi; curtain gas (CUR), 35 psi; temperature (TEM), 600 °C; ion spray voltage floating (ISVF), 5000 V or −4000 V, in positive or negative modes, respectively; declustering potential (DP), 60 V or −60 V in positive or negative modes, respectively. LC–MS data acquisition was operated under information-dependent acquisition (IDA) mode. The instrument was set to acquire over the $m/z$ range 60–1200 Da for TOF MS scan and the $m/z$ range 25–1200 Da for product ion scan. Collision energy for product ion scan was 30 eV or −30 eV in positive or negative modes, respectively, while collision energy spread (CES) was 0 eV. The accumulation time for TOF MS scan was set at 150 ms per spectrum and product ion scan at 30 ms per spectrum. The unit resolution was selected for precursor ion selection. IDA settings were set as followings: charge state 1 to 1, intensity 100 cps, exclude isotopes within 4 Da, mass tolerance 10 ppm, and maximum number of candidate ions 12. The "exclude former target ions" was set as 4 s after two occurrences. In IDA advanced tab, "dynamic background subtraction" was chosen.

**Data processing**. The raw data was acquired from orbitrap mass spectrometer using Xcalibur (version 4.4.16.14, Thermo Fisher Scientific, USA) and TOF mass spectrometer using Analyst TF Software (version 1.7.1, Sciex, USA). For data acquired from the orbitrap mass spectrometer, the raw MS data files (.raw) were converted to.mzXML (for full scan mode) and.mgf (for ddMS2 mode) format using ProteoWizard (version 3.0.20360). Then, mzXML data files of unlabeled samples were grouped for peak detection and alignment using the R package "xcms" (version 3.12.0; https://bioconductor.org/packages/release/bioc/html/xcms.html). Key parameters were set as follows: method, "centWave"; ppm, 10; snthr, 10; peakwidth, c(5,30); minfrac, 0.5. For data acquired using TOF mass spectrometer, the raw MS data files (.wiff) were converted to.mzXML format using ProteoWizard (Version 3.0.6150). Then, mzXML data files of unlabeled samples were grouped for peak detection and alignment using R package "xcms" (version 1.46.0; https://bioconductor.org/packages/release/bioc/html/xcms.html). The key parameters were set as follows: method, "centWave"; ppm, 25; snthr, 10; peakwidth, c(5,30); minfrac, 0.5. The intermediate xcmsSet object from "xcms" after peak detection was exported as a xcmsSet file (.Rda). The generated MS1 peak table and MS2 files were uploaded to MetDNA[64] (version 1.2.2; http://metdna.zhulab.cn/) for metabolite annotation. The metabolite annotation parameters were set as "HILIC" or "RP" according to liquid chromatography mode, "Sciex TripleTOF" or "Thermo Exploris" according to instrument platform, and "30" or "SNCE_20_30_40%" for collision energy. We performed the metabolite annotation separately on both positive and negative mode. Before MetTracer analysis, MetDNA annotation results were further filtered, including removal of isotope annotations, removal of grade 4 annotations, removal of redundant peaks, while annotation types such as seed and metAnnotation, and annotations with reliable adducts (e.g., $[M + H]^+$, $[M + Na]^+$, $[M + NH4]^+$, $[M-H]^-$, $[M + Cl]^-$, $[M-H-H_2O]^-$) were kept. Finally, MS-Finder (version 3.24; http://prime.psc.riken.jp/Metabolomics_Software/MS-FINDER/index.html) was used for formula prediction (top 5 candidates) and annotation filter. The final annotation table for 293T cells and fly samples were provided in Supplementary Data 1–3. According to the definition of metabolomics standards initiative (MSI), we assigned the metabolite annotations with three confidence levels. Level 1 means metabolites annotated through matching of MS1, RT and MS/MS spectra with the in-house metabolite spectral library. Level 2 means metabolites annotated through matching MS1 and MS/MS2 spec with public metabolite spectral library (mainly from NIST 2017). Level 3 means metabolites annotated based on MS1 and surrogated MS/MS match using MetDNA. In total, 1,035 metabolites were annotated in 293T cell samples acquired on Orbitrap Exploris 480, including 347, 144, and 544 metabolites with annotation MSI level 1, 2, and 3, respectively. For 293T cells samples acquired on TripleTOF 6600, a total of 1347 metabolites were annotated, including 215, 219, and 913 metabolites with annotation MSI level 1, 2, and 3, respectively. For fly head samples, a total of 745 metabolites were annotated, and 112, 91, and 542 metabolites were annotated as MSI level 1, 2, and 3, respectively. For fly muscle samples, a total of 1290 metabolites were annotated, and 117, 102 and 1071 metabolites were annotated as MSI

level 1, 2, and 3, respectively. In addition to MetDNA, MetTracer also supports metabolite annotation results from other software tools such as MS-DIAL[65], SIRIUS CSI-FingerID[66], and GNPS[67].

For MetTracer analysis, the metabolite annotation table (Supplementary Data 1–3), the previously generated xcmsSet file from XCMS, and raw data files (.mzXML) from unlabeled samples were organized in a folder named "unlabeled". The raw data files (.mzXML) from labeled samples were organized in a folder named "labeled", and separated into several subfolders according to their experimental groups. Finally, the "unlabeled" and "labeled" folders were subjected to the R package "MetTracer" for global tracking of labeled metabolites. The parameters for MetTracer were set as follows: rt.extend, 15; value, "maxo"; d.extract, "labelled"; correct.iso, "TRUE"; adj.contaminate, "TRUE". The detailed parameter setting for MetTracer is provided in Supplementary Table 8. In labeled samples, if the labeled fraction of one isotopologue (except M0) in the metabolite is larger than 0.02 in >50% of samples, we consider the metabolite was isotopically labeled.

**The workflow of MetTracer**. The MetTracer data processing workflow includes three major steps: (1) generation of a targeted list for isotopologues; (2) extraction of isotopologue peaks, and (3) correction and quantification.

(1) Generation of isotopologue targeted list. The metabolite formulas in the annotation table were used to calculate the theoretical $m/z$ values of all possible $^{13}$C-isotopologues (M0-Mn) for each metabolite with the R package "enviPat"[68] (version 2.4). For each monoisotopic peak (M0), the corresponding extracted ion chromatogram (EIC) peaks were extracted from each unlabeled sample according to the feature information recorded in the xcmsSet file. The EIC peak with highest peak height was selected as the "targeted EIC" to aid the extraction of labeled metabolites in labeled samples. The retention time and peak shape of the targeted EIC were also recorded in the isotopologue targeted list. Finally, for each annotated metabolite, the isotopologue targeted list included $m/z$ values of all possible $^{13}$C-isotopologues, retention time, and peak shape of the target EIC.

(2) Extraction of isotopologue peaks. The isotopologue targeted list was used for targeted extraction of isotopologue peaks in the labeled samples (.mzXML). For each metabolite, the ion chromatograms of all isotopologues were extracted within the extended retention time (RT) range ([RT$_{\text{left boundary}}$ − 15 s, RT$_{\text{right boundary}}$ + 15 s]). The $m/z$ tolerance of extraction was set as 25 ppm (or 0.01 Da for ion <400 Da). For each EIC of isotopologue, the noise level and baseline were determined. The EIC with maximum signal-to-noise (S/N) ratio lower than 3 were removed. Then the EICs were smoothed using "Gaussian" method and subjected to peak detection with the "centWave" algorithm. The peak-peak correlation (PPC) was also calculated to characterize the similarity between the detected isotopologue EIC peaks and the targeted EIC in the isotopologue targeted list using a modified Pearson correlation coefficient. All isotopologue peaks with PPC ≥ 0.6 were kept. For the remaining isotopologue peaks, hierarchical-clustering analysis was applied for clustering the peak apexes with a cutoff threshold of 3 s, which indicated that the RT differences among peak apexes should be <3 s in one isotopologue peak group. The isotopologue peak group with the closest RT to the targeted EIC was selected. The RT range and peak apex of isotopologue peaks in the selected isotopologue peak group was re-adjusted according to the peak with highest peak height in the group.

(3) Correction and quantification. For each isotopologue peak of one metabolite, the sum intensity of three scans around the peak apex was used as a proxy to the intensity of the peak. If one isotopologue in isotopologue targeted list was not detected, this isotopologue peak was extracted mandatorily according to the theoretical $m/z$ and RT range. Next, natural isotope correction was performed using the R package "AccuCor" (version 0.2.4; https://github.com/XiaoyangSu/AccuCor) to obtain the corrected peak intensities and mass isotopomer distribution (MID) table. The MID describes the labeled fraction of each isotopologue of metabolites. Then the isotope contamination was estimated in unlabeled samples. For each metabolite, if monoisotopic peak (M0) intensities were 0 in >50% of unlabeled samples, we considered that the metabolite did not exist in unlabeled samples. In labeled samples, the labeled fractions of all the isotopologues of this metabolite were set as 0. In unlabeled samples, if the labeled fraction of one isotopologue (except M0) is larger than 0.02 in >50% of samples, we consider the isotopologue was isotopically contaminated. We assumed that the contamination levels in both labeled and unlabeled samples were the same. Thus, the labeled fractions of contaminated isotopologues in labeled samples were corrected by subtracting the contaminate level estimated in unlabeled samples. Finally, the MID table that describes the labeled fractions of isotopologues of each metabolite was outputted as the final result.

**Labeling extent (LE) calculation**. Labeling extent (LE) represents the labeling enrichment of one metabolite[69]. It is calculated by Eq. (1):

$$\text{LE} = \frac{\sum_{i=1}^{C} I_{\text{Mi}}}{\sum_{i=0}^{C} I_{\text{Mi}}} = 1 - \frac{I_{M0}}{\sum_{i=0}^{C} I_{\text{Mi}}} = 1 - L_{M0} \tag{1}$$

where $I_{Mi}$ is the peak intensity of isotopologue Mi, $L_{M0} (= \frac{I_{M0}}{\sum_{i=0}^{C} I_{Mi}})$ is the labeled fraction of $M0$, $C$ is the total carbon number of the metabolite. Moreover, the

labeling extent of a given metabolic pathway in Fig. 3C is represented by the mean labeling extent of all labeled metabolites in the pathway.

**Labeling rate calculation**. Here, in global isotope tracing analysis, we fitted the in vivo tracing kinetics of the isotopically labeled metabolites using the general first-order exponential equation for large-scale quantitation of the labeling rates of metabolties, which correspond to the incorption rate of the tracer to the downstream metabolite targets. Labeling extents (LE) for each metabolite at all time points were used to fit the exponential function.

$$LE = a \times e^{(-kt)} - a \, (a < 0) \qquad (2)$$

$k$ and $a$ are fitted using R package "nls" from the LE values, where $k$ is the first-order rate constant, indicating the incorporation rate of the tracer to a metabolite target. Moreover, the labeling rate of a given metabolic pathway in Fig. 2 is represented by the median labeling rate of all labeled metabolites in the pathway.

**Benchmark of coverage performance**. The extractions of labeled metabolites using X[13]CMS[20], geoRge[22] and El-MAVEN[29] were evaluated as follows. For X[13]CMS, the R package "X13CMS" was downloaded from the website (version 1.4; http://pattilab.wustl.edu/software/x13cms/x13cms.php). For geoRge, the R package "geoRge" was downloaded from GitHub (version 1.0; https://github.com/jcapelladesto/geoRge). Both X[13]CMS and geoRge provided the result table of [13]C-labeled isotopologues for each peak group. Then, the unlabeled peak in the peak group was matched with the peak in MetDNA annotation table according to the *m/z* and RT. For El-MAVEN, the GUI software was used(version 0.11.0; https://resources.elucidata.io/elmaven). The list of 1347 annotated metabolites including metabolite name, formula, and retention time and the raw data files (.mzXML) were imported into El-MAVEN for targeted extraction of labeled metabolites. In El-MAVEN, the labeled metabolite was defined if any isotopologue except M0 had the labeled fraction >0.02 in >50% of samples. The detailed parameters of these software tools were provided in Supplementary Table 9. The main parameters modified of these three softwares and the reason for modification were provided in Supplementary Table 10.

**Manual analysis of labeled metabolites using Skyline**. The Skyline software (version 20.2.0.286; https://skyline.ms/project/home/software/Skyline/begin.view) was used for manual analysis of labeled metabolites. The raw data files (.mzXML) and the isotopologue target list were imported into Skyline. For each isotopologue, the integration range was manually checked and adjusted for accurate quantification of each isotopologue peak. The quantification result was also corrected with "Accucor" for natural isotope correction, and then labeled fractions and labeling extents were calculated. The relative errors for labeled metabolites were calculated by Eq. (3):

$$\text{Relative error} = \frac{\text{LE(MetTracer)} - \text{LE(Skyline)}}{\text{LE(Skyline)}} \qquad (3)$$

**False-positive rate evaluation**. First, both unlabeled and labeled samples were analyzed by MetTracer and El-MAVEN using the above protocol. Next, in the unlabeled samples (instead of labled samples), the labeled fraction of isotopologues were calculated. We defined the isotopologue with the labeled fraction >2% in any one sample as a false positive. False-positive rate at isotopologue level referred to the ratio of the number of false-positive labeled isotopologues to the number of all isotopologues for all metabolites. False-positive rate at metabolite level referred to the ratio of the number of labeled metabolites to the number of all extracted metabolites. Detailed numbers were listed in Supplementary Tables 2–4. There is a technical limitation to estimate false-positive rate in untargeted methods such as X[13]CMS and geoRge. They require both unlabeled and labeled samples as inputs to determine the labeled isotopologues and metabolites. The unlabeled samples were necessary in the data processing and served as control samples. As a result, it is not possible to estimate the false-positive rates of X[13]CMS and geoRge using the similar method as MetTracer and El-MAVEN.

**Metabolic pathway-enrichment analysis**. Metabolic pathway-enrichment analysis was performed via hypergeometric test[70]. For the enrichment analysis of the labeled metabolites in cluster 1, 2, and 3 in the head and muscle tissues of *Drosophila*, the background database was all of the labeled metabolites in the head and muscle tissues, respectively. For the enrichment analysis of the shared labeled metabolites in both head and muscle tissues, the background database was the metabolites in KEGG.

**Hierarchical-clustering analysis**. Hierarchical-clustering was performed using an R package "pheatmap" (version 1.0.12) with the default parameters. The mean labeling extent values of metabolites in each labeling time point were used. The clustering method was "ward.D" for head and "complete" for muscle. The correlation values were used as the clustering distance in heat map plot.

**Metabolite–metabolite correlation calculation**. The labeling extent values after 24 h labeling were used for Pearson correlation calculation between metabolites from each cluster. Only metabolite correlations with *p*-value <0.05 after FDR correction were retained to construct the correlation network. The correlation network was visualized by R package "circlize" (version 0.2.10).

**PolyA-selected RNA-seq**. For RNA-seq experiments, dissected fly head tissues (3d and 30d) were used. Tissues were homogenized in a 1.5 mL tube containing 1 mL of Trizol Reagent (Thermo Fisher Scientific, USA). RNA isolation was followed in accordance with manufacturer's instruction. RNA was resuspended in DEPC-treated RNase-free water (Thermo Fisher Scientific). TURBO DNA free kit was used to remove residual DNA contamination according to manufacturer's instruction (Thermo Fisher Scientific). One microgram of total RNA was used for sequencing library preparation. PolyA-tailed RNAs were selected by NEBNext Poly(A) mRNA Magnetic Isolation Module (New England Biolabs, USA), followed by the library preparation using NEBNext Ultra RNA library Prep Kit for Illumina according to manufacturer's instruction (New England Biolabs, USA). Libraries were pooled and sequenced on the Illumina Miseq platform with single end 100 bps (Illumina, USA). Sequencing reads were mapped to the reference genome dm6 with STAR2.3.0e by default parameter. The read counts for each gene were calculated by HTSeq-0.5.4e htseq-count with parameters "-m intersection-strict -s no" with STAR generated SAM files. The count files were used as input to R package "DESeq" (version 1.8.3) for normalization.

**Reporting summary**. Further information on research design is available in the Nature Research Reporting Summary linked to this article.

## Data availability

The raw metabolomics and RNA-seq data files generated in this study have been deposited in the National Omics Data Encyclopedia under accession code OEP002699. The raw metabolomics data files have also been deposited in Metabolights under accession code MTBLS3322. The RNA-seq data files generated in this study have been deposited in the Gene Expression Omnibus database under accession code GSE204740. Sequencing reads were mapped to the *Drosophila Melanogaster* reference genome dm6. The annotation results for all metabolomics datasets are provided in the Supplementary Data 1–3. Source data are provided with this paper.

## Code availability

The source code of MetTracer is provided on GitHub https://github.com/ZhuMetLab/MetTracer and Zenodo https://doi.org/10.5281/zenodo.6575308[71].

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

## Acknowledgements

The work is financially supported by National Natural Science Foundation of China (92057114 and 22022411 to Z.J.Z.), National Key R&D Program of China (2018YFA0800902 to Z.J.Z.), Shanghai Municipal Science and Technology Major Project (2019SHZDZX02 to Z.J.Z.), and Science and Technology Commission of Shanghai Municipality (21JC1405902 to Z.J.Z.).

## Author contributions

Z.J.Z. and R.W. conceived the idea and designed the project. R.W. performed the sample preparation, data acquisition, and data analysis. Y.Y. contributed to the software. H.W., W.L., Y.G., and Z.Z. contributed to the data analysis. J.L., T.W., Y.Z., and N.L. contributed to the fly culture and dissection. X.S. contributed to the isotope correction. Z.J.Z., Y.C., and R.W. wrote the manuscript. Z.J.Z. supervised the project.

## Competing interests

The authors declare no competing interests.

## Additional information

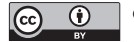 ns license, unless indicated otherwise in a credit line to the material. If material is not included in the article's Creative Commons license and your intended use is not permitted by statutory regulation or exceeds the permitted use, you will need to obtain permission directly from the copyright holder. To view a copy of this license, visit http://creativecommons.org/licenses/by/4.0/.

© The Author(s) 2022

