## [Peer Review File · Nature Communications]

REVIEWER COMMENTS

Reviewer #1 (Remarks to the Author):

In this manuscript, Wang et. al describe the development of an untargeted metabolomics technique with the focus of measuring a large range of ¹³C-labeled metabolites and apply the technique to an in vivo model, namely *Drosophila melanogaster*. Authors also develop a kinetic flux profiling method to quantify in vivo labeling and apply these techniques to measure differences in ¹³C glucose labeling in fly head and muscle, and in young and aged flies. Authors conclude that there is a “system-wide loss of metabolic coordinations that impacts both intra- and inter-tissue metabolic homeostasis significantly during *Drosophila* aging” and that there is decreased glycolysis and increased serine and one-carbon metabolism in aged flies. While these techniques and findings are interesting, there are some major concerns about the validity of using kinetic flux profiling in this scenario.

Major:

The kinetic flux profiling (KFP) has limitations. First, the idea of KFP requires a fast switch from unlabeled substrate to labeled substrate, and hence the upstream source can be regarded as constant. As shown in Jie Yuan et al, 2008 on Nature Protocol, this method is developed with cultured cells, of which labeling substrate in media can be rapidly exchanged. However, in the situation with fruit flies, the labeling by diet is slow: flies need to eat, digest, absorb ¹³C substrate and distribute it to the whole body. All those steps are much slower than that in cultured cells. Under this condition, the concise exponential decay equation may not be valid, and the fitting may not be meaningful. The authors should provide more evidence to prove that this method is still a good approximation for ¹³C labeling by diet in *Drosophila*.

Similar with 1, since KFP requires a fast switch of upstream source than the downstream sink, the best situation to use it is the rate-limiting reaction from a pathway with fast and large flux to a pathway with slow and small flux. For example, glucose intake fluxes are usually much faster than glycolysis and TCA cycle, and then the hexokinase will be the best reaction for KFP. Similarly, glycolysis and TCA cycle are faster than branching pathways such as one carbon metabolism, base synthesis, or fatty acid synthesis, thus their rate-limiting reactions are good for KFP. However, the author seem to use KFP for all reactions. They do not provide any detail information on the metabolic network they use, either. It will be hard to evaluate whether they use KFP in correct way.

The term of “inter-tissue metabolic activities” used in the section “Quantitative stable-isotope tracing reveals distinct inter-tissue metabolic activities” may not be appropriate enough. According to my understanding, this term means direct interaction in metabolism between different kinds of

tissue, but the author only shows distinct metabolic profiles in different tissue, and no evidence for direct interaction is provided. My suggestion is to change it to some words like “distinct metabolic profiles in different tissue”.

Minor:

In figure 2d, I suggest to either explicitly mark $a < 0$, or let $a = -a$ and use the opposite form $a - a \times e^{(-kt)}$. This will make the equation more intuitive for readers.

More details on the fly dissection process should be given, especially noting:

Were the flies dissected under CO₂ anesthesia?

How long did the dissection process take before the tissues were flash-frozen?

What type of muscle tissue was collected from flies? Was the whole head used or was a certain part of the head dissected out?

Reviewer #2 (Remarks to the Author):

The paper by Wang et al. reports development of a tool, named MetTracer, for tracing labeled metabolites in stable-isotope feeding experiments. The authors show that MetTracer is very capable and permits extraction of an impressive number of isotopologues of annotated metabolites. Figure 1 (and some SI Figures) compare MetTracer with results obtained using other tools (e.g. X13CMS). Such comparisons are difficult to judge, because output of each of the tools being compared may depend significantly on the choice of tool-specific parameters. Detailed parameters used for each of the tools are provided in Supplementary Table 7, which is nice, but it is unclear whether these were optimized to ensure best performance of each of the tools.

After demonstrating the capabilities of MetTracer in a cell culture system, the manuscript presents application to *Drosophila*. The authors show that different groups of pathways incorporate labels at different rates, and that labeling follows a roughly exponential model, as can be calculated based on the output from MetTracer. Further, it is shown that head and muscle feature different labeling profiles and that, as *Drosophila* ages, there is a shift away from glycolysis toward serine and purine metabolism. These findings are well supported by the data, the results are in line with expectations based on previous work by the authors and others, and the coverage achieved with MetTracer is impressive.

In addition, the authors claim that their data revealed disclose a “system wide loss of metabolic coordination that impacts both intra and inter tissue metabolic homeostasis significantly”. This statement appears to be based on the fact that Pearson correlation for different metabolites becomes much less significant for the 30-day-old cohort compared to the 3-day cohort. However, for this analysis, it is essential to consider that, per se, observation of aged animals results in increased heterogeneity, because individuals age at different rates (correspondingly, not all animals die at the same time). As far as I can see, this is not explicitly considered. Therefore, I do not think the parts of the manuscript discussing “loss of metabolic coordination” are supported by the data. As a result, the manuscript may not be appropriate for publication in its present form, though MetTracer appears to be a very useful tool for stable isotope labeling studies.

Minor points:

Figure 1: there are a few mistakes, e.g. "Correction for natural isotope..." and "isotopomer" in panel b.

Line 125: explicitly state that this is the in-silico calculated number of isotopologues, of which then a subset is detected. This entire section is difficult to read and could benefit from clarification, especially with regard to what actually is done by MetTracer (the Methods section was somewhat helpful, but the main text section is nearly incomprehensible as is).

Line 127: It is unclear what is improved - recognition of peaks, or assignment as isotopologues of a specific metabolite.

Line 133: it is unclear what "good quantitative consistency" means, relative to the statement in the previous sentence.

Line 139: Again, it's unclear whether the better performance of MetTracer is due to better peak recognitions (i.e. fewer dropouts) when using MetTracer.

Line 168: “was the fastest labeling group” - isn't this simply the result of the criteria used for clustering?

Line 185: “Further, ...” This again merely restates that labeling rates were found to be different for different metabolite groups, i.e. the different clusters correspond to metabolites from different pathways.

Reviewer #3 (Remarks to the Author):

Wang et al. describe MetTracer, a novel system for tracing metabolites across flies of different ages to gain insights into how head and muscle metabolomes alter with age. The analyses utilized for this study provide clear results to demonstrate the utility of MetTracer; however, there are a number of concerns that must be addressed for this paper to be publishable.

1) The study used a w1118 fly model. While this strain is commonly used as a 'wild type' of sorts, it notably has a mutation to a gene that influences purine transport, neurotransmitter function, and eye function. There is a strong likelihood that this mutation alters the metabolome, thus skewing the results presented in this study. While this does not hurt the main point the authors are making when demonstrating the utility of MetTracer, it does provide unintended context to the dataset they generated, especially considering how strongly purine metabolism was represented in the dataset. It would be worth discussing this and adding appropriate references.

2) In Figure 1f, I wonder why the comparison in reproducibility and false positives is restricted to just being between MetTracer and EI-MAVEN. Considering EI-MAVEN was the tool which had the least overlap with MetTracer (presented in Figure 1d), I think it would be useful to compare its reproducibility to other tools which have more similar overlap in signatures (or better yet, a comparison between all tools presented in 1d).

3) While the tool described is novel, age-related metabolomic analyses in *Drosophila* have been performed many times (Coquin et al 2008, Avanesov et al 2014, Hoffman et al 2014, Laye et al 2014, Zhou et al 2017, Jin et al 2020, Parkhitko et al 2020, Tapia et al 2021, Hall et al 2021, Zhao et al, 2022). A comparison between the results from these studies and the new approach presented in this manuscript would help demonstrate the value of the authors' method. Depending on the fly strains used in the other studies, this could also alleviate the concerns about the w1118 model or affirm the results from this study.

Minor concerns:

4) There are a number of typographical errors:

- In line 35, 'facilitates' is spelled incorrectly.
- Line 155 has an unnecessary space before the comma.
- The sentence in lines 227-229 is incomplete and probably should be combined with the preceding sentence.
- Labeling of ages is not consistent throughout the manuscript (e.g. '30 d' in line 324 vs '30-d' in line 328 vs '30d' in the figures)
- In line 388, 'age-dependent' is misspelled.

- In line 442, 'wild-type' is misspelled as 'wide type.'

5) Why were head and muscle chosen as the tissues of interest?

6) Figure 4 requires some additional descriptive details. How were the metabolites within each cluster in the Circos plots arranged? And how are inter-cluster correlations affected with age?

7) The individual datapoints in some of the box plots are too small to make out, especially in Figure 5d. If the size of these could be increased, it would be helpful.

8) How was gene expression determined in Figure 5? This should be briefly mentioned in the text and detailed in the Methods.

9) Various points in the manuscript (like lines 265 and 335) mention a "previous report" but do not provide a citation.

10) For the described PRC2 mutation, is this referring to mutations to genes in the Polycomb repressive complex 2? This needs to be clarified.

11) Reference to Yuan et al. in the discussion (line 376) requires a citation.

12) Some of the Discussion section is redundant (e.g. lines 396 and 409 repeat each other).

13) Figure 4/results- typo in subplot c legend, liner regression should be linear regression

14) What is purpose of 5a? What are the non-labeled nodes?

15) Same with 5f

A model, but only thing that changes is thickness of roundabout and the seesaw

Response to the reviewers:

The authors would like to thank the reviewers for the helpful comments. We feel these comments have strengthened the manuscript considerably.

Reviewer #1

Remark to author: *“In this manuscript, Wang et. al describe the development of an untargeted metabolomics technique with the focus of measuring a large range of ¹³C-labeled metabolites and apply the technique to an in vivo model, namely *Drosophila melanogaster*. Authors also develop a kinetic flux profiling method to quantify in vivo labeling and apply these techniques to measure differences in ¹³C glucose labeling in fly head and muscle, and in young and aged flies. Authors conclude that there is a “system-wide loss of metabolic coordinations that impacts both intra- and inter-tissue metabolic homeostasis significantly during *Drosophila* aging” and that there is decreased glycolysis and increased serine and one-carbon metabolism in aged flies. While these techniques and findings are interesting, there are some major concerns about the validity of using kinetic flux profiling in this scenario.”*

Ans: We appreciate the comments from the Reviewer. We agreed with the Reviewer’s concern about the validity of kinetic flux profiling, and thoroughly revised the manuscript to address these comments. We feel these comments have significantly strengthened the manuscript.

Comment #1: *“The kinetic flux profiling (KFP) has limitations. First, the idea of KFP requires a fast switch from unlabeled substrate to labeled substrate, and hence the upstream source can be regarded as constant. As shown in Jie Yuan et al, 2008 on Nature Protocol, this method is developed with cultured cells, of which labeling substrate in media can be rapidly exchanged. However, in the situation with fruit flies, the labeling by diet is slow: flies need to eat, digest, absorb ¹³C substrate and distribute it to the whole body. All those steps are much slower than that in cultured cells. Under this condition, the concise exponential decay equation may not be valid, and the fitting may not be meaningful. The authors should provide more evidence to prove that this method is still a good approximation for ¹³C labeling by diet in *Drosophila*.”*

Ans: Thanks a lot for the Reviewer's comment. We also had the similar concern about the kinetic flux profiling (KFP) model. When we were preparing the manuscript, we were also struggling on this point for a while. We know that the KFP model was often used to calculate the **metabolic flux** by fitting the first-order rate constant k . Then, the metabolic flux was calculated through multiplying the rate constant k by metabolite concentration. We agree with the Reviewer that the validity of KFP model requires the rapid switch of labeling substrate. In our work, we understand that KFP model for metabolic flux calculation may not be valid for *in vivo* isotope tracing of *Drosophila*.

However, through the data fitting, we did observe that *in vivo* tracing kinetics of *Drosophila* over 24 h conformed **the first-order exponential equation** which is similar to the KFP model (see fitting

examples in **Supplementary Figure 9 and 15**). Our data showed that, in the head tissue, 75% (292 out of 390) metabolites had *in vivo* tracing kinetics successfully fitted with a first-order exponential equation. Similarly, in the muscle tissue, 73% (437 out of 597) metabolites had successful fittings. Therefore, the results indicated that the majority of metabolites in *Drosophila* were metabolized following the first-order exponential function. **The labeling rate k indicated the incorporation rate of [U-¹³C]-glucose tracer to the downstream metabolites.**

Some of previous reports had used the first-order exponential equation to fit the labeling kinetics and calculate the labeling rates of proteins and metabolites (Hammond *et al.*, Molecular & Cellular Proteomics, **2018**; <https://doi.org/10.1074/mcp.TIR117.000516>; Schlame *et al.*, Journal of Lipid Research, **2020**; <https://doi.org/10.1194/jlr.D119000318>; Shi *et al.* Analytical Chemistry, **2020**; <https://doi.org/10.1021/acs.analchem.0c01767>). In particular for lipid analysis in adult *Drosophila*, Schlame *et al.* used the first-order exponential equation to calculate the turnover rate k , and determined the flux by multiplying the k value of a lipid with its concentration. **However, according to the Reviewer's comment, we agree that we should NOT simply define the *in vivo* tracing kinetics of *Drosophila* as the conventional kinetic flux profiling (KFP) model, which may cause misunderstandings to the readers.** Most importantly, it is **NOT** valid to calculate metabolic fluxes through multiplying k values by metabolite concentrations in our results since the assumption of rapid switch of labeling substrate was not satisfied. Therefore, in the revised manuscript, we thoroughly removed the statements about the KFP model in the whole manuscript, and revised as "*in vivo* tracing kinetics" or "*the first-order exponential equation*". For example, the related description of **Figure 2** in manuscript has been revised as:

"We then fitted the in vivo tracing kinetics of labeled metabolites in Drosophila using the general first-order exponential function, and enabled large-scale quantitation of labeling rates from [U-¹³C]-glucose tracer to the downstream metabolites (see Methods). Labeling extents for each metabolite at all time points were used to fit the exponential function. Thus, the labeling rate k , which is the apparent first-order constant in the exponential function, was calculated to quantify the incorporation rate of the tracer to metabolite targets in the continuous stable-isotope labeling.

We have also added the related points in Discussion in the revised manuscript:

"Previously, Yuan et. al. developed the kinetic flux profiling (KFP) analysis for metabolic flux calculation which required a fast switch of labeling substrates. As such, the KFP model is typically used in cultured cells wherein the labeling substrate in media can be rapidly exchanged. Here, the KFP model for metabolic flux calculation may not be valid for in vivo isotope tracing of Drosophila. Instead, we fitted the in vivo tracing kinetics of labeled metabolite using the first-order exponential equation to calculate labeling rate k value. The labeling rate k quantifies the incorporation rate of the [U-¹³C]-glucose tracer to downstream metabolite targets in the continuous stable-isotope labeling. Our data showed that, in the head tissue, 75% metabolites (292 out of 390) had the in vivo tracing kinetics successfully fitted with the first-order exponential equation. Similarly, 73% metabolites (437 out of 597) in the muscle tissue had successful fittings. Since the k value indicates the incorporation rate of the tracer to a metabolite target, it is reasonable that the isotopically labeled metabolites nearer

to the [U-¹³C]-glucose tracer in the metabolic network have higher k values being calculated. Indeed, we showed that metabolites in glycolysis such as pyruvate exhibited larger k values than those in fatty acid synthesis such as hexadecanoic acid (**Supplementary Figure 15**). Of note, we cannot determine the absolute metabolic flux through multiplying the k values calculated here by metabolite concentrations because the assumption of KFP analysis is not met in *Drosophila*. This limitation, however, does not detract from our fittings of *in vivo* tracing kinetics on a metabolome-wide scale and comparative analyses of k values for the same metabolite between different biological scenarios.”

Supplementary Figure 15. Examples of metabolites with different labeling rates in head tissue of *Drosophila*. Data are presented as median \pm SD; $n = 10$ biological replicates in each time point; grey line, fitting curve.

Comment #2: “Similar with 1, since KFP requires a fast switch of upstream source than the downstream sink, the best situation to use it is the rate-limiting reaction from a pathway with fast and large flux to a pathway with slow and small flux. For example, glucose intake fluxes are usually much faster than glycolysis and TCA cycle, and then the hexokinase will be the best reaction for KFP. Similarly, glycolysis and TCA cycle are faster than branching pathways such as one carbon metabolism, base synthesis, or fatty acid synthesis, thus their rate-limiting reactions are good for KFP. However, the author seem to use KFP for all reactions. They do not provide any detail information on the metabolic network they use, either. It will be hard to evaluate whether they use KFP in correct way.”

Ans: Thanks a lot for the Reviewer’s comments toward the proper use of KFP model for metabolic flux calculation. We agree that the accurate calculation of metabolic fluxes using the KFP model should consider the rate-limiting reaction from a pathway with fast and large flux to a pathway with slow and small flux. Again, the requirement should be applied to the metabolic flux calculation using KFP. In our study, we think that **it is not valid** to calculate metabolic fluxes by multiplying the k values by metabolite concentrations because the assumption of KFP analysis is not satisfied in *Drosophila*. The k values only indicated the labeling rates from [U-¹³C]-glucose tracer to the labeled metabolites. Therefore, it is reasonable that the isotopically labeled metabolites nearer to the [U-¹³C]-glucose tracer in the metabolic network have higher k values being calculated. As shown in Figure 2f, the mean

labeling rate k values were gradually decreased from carbohydrate metabolism, amino acid metabolism to fatty acid metabolism. The results were consistent with the fact that [U-¹³C]-glucose was used as the tracer in the study. Biological conversion from glucose to fatty acids took more metabolic reactions and a longer time. Therefore, fatty acid metabolism had a slower labeling rate k . More specifically, in **Supplementary Figure 15**, the labeling rates of pyruvate, aspartate, and hexadecenoic acid were decreased according their metabolic distances from the [U-¹³C]-glucose tracer.

Most importantly, we restricted the use of labeling rate k values, which could only be used for comparison of the same metabolite under different biological conditions (such as young vs. old *Drosophila*). In such a scenario, we may not have to consider the requirement of the rate-limiting reactions as KFP model. **Finally, we also wish the Reviewer could understand that the labeling rate k calculation was a small part of our manuscript. In our study, the major biological conclusions in Figures 3, 4 and 5 were mainly obtained using the labeling extent (LE) and labeled fractions of isotopologues, which had no doubt on accuracy.**

Comment #3: *“The term of “inter-tissue metabolic activities” used in the section “Quantitative stable-isotope tracing reveals distinct inter-tissue metabolic activities” may not be appropriate enough. According to my understanding, this term means direct interaction in metabolism between different kinds of tissue, but the author only shows distinct metabolic profiles in different tissue, and no evidence for direct interaction is provided. My suggestion is to change it to some words like “distinct metabolic profiles in different tissue”.”*

Ans: Thanks a lot for the Reviewer’s comment. We agreed to the comment and the sentence has been revised as *“Quantitative stable-isotope tracing reveals distinct metabolic profiles in different tissues”* in the revised manuscript.

Comment #4: *“In figure 2d, I suggest to either explicitly mark $a < 0$, or let $a = -a$ and use the opposite form $a - a \times e^{(-kt)}$. This will make the equation more intuitive for readers.”*

Ans: Thanks for the Reviewer’s comment. We have marked $a < 0$ in the equation in the revised manuscript.

Comment #5: *“More details on the fly dissection process should be given, especially noting: Were the flies dissected under CO₂ anesthesia? How long did the dissection process take before the tissues were flash-frozen? What type of muscle tissue was collected from flies? Was the whole head used or was a certain part of the head dissected out?”*

Ans: Thanks a lot for the Reviewer’s comment. We have provided the details in the “Methods” section in the revised manuscript. The flies were dissected under CO₂ anesthesia. It took about 15-20 seconds to dissect a fly. After dissection, tissue samples were quickly frozen using liquid nitrogen. The whole muscle tissues and whole head tissues were collected from flies and used for the LC-MS experiments.

Reviewer #2

Comment #1: “The paper by Wang et al. reports development of a tool, named MetTracer, for tracing labeled metabolites in stable-isotope feeding experiments. The authors show that MetTracer is very capable and permits extraction of an impressive number of isotopologues of annotated metabolites. Figure1 (and some SI Figures) compare MetTracer with results obtained using other tools (e.g. X13CMS). Such comparisons are difficult to judge, because output of each of the tools being compared may depend significantly on the choice of tool-specific parameters. Detailed parameters used for each of the tools are provided in Supplementary Table 7, which is nice, but it is unclear whether these were optimized to ensure best performance of each of the tools.”

Ans: Thanks for the Reviewer’s comment. In the manuscript, we demonstrated that MetTracer is a high-coverage isotope tracing metabolomics technology with high reproducibility and quantification accuracy, as well as low false positive rate. To make a fair comparison, we have implemented the following procedures:

- 1) We have used the same dataset as the input for different software tools;
- 2) To minimize potential bias from different instrument platforms, we performed the comparisons on both TOF (TripleTOF 6600) and Orbitrap (Exploris480) instruments;
- 3) We comprehensively conducted comparisons from three different aspects of each software tool, including **a)** coverages of metabolites and isotopologues; **b)** quantification reproducibility; **c)** false positive rate;
- 4) As there are no specific tools to optimize the parameters, to make the comparison as fair as possible, we carefully optimized the parameters for each software tool according to the instrument platforms and **similar parameters were also used in MetTracer**. In **Supplementary Table 10** of the revised manuscript, we have provided the default values, modified values of the main parameters in each software tool. In addition, reasons for modification were also explained.

With these implementations, we think that we have tried our best to provide fair comparisons between different software tools. We also agree with the Reviewer that it is always not easy to judge the comparisons. Therefore, we also added the following statements in the revised manuscript,

“Although we have tried our best to perform fair comparisons between different software tools, and provided the detailed parameters for each tool, such comparisons are still difficult to judge, because output of each tool being compared may depend significantly on the choice of tool-specific parameters”.

Supplementary Table 10. The modifications of default parameters in different software tools.

Software	Parameter	Default	Modified	Reason
	RTwindow	NA	3 s	Keep the same as MetTracer.
	ppm	NA	25 ppm (QTOF)	Keep the same as MetTracer.

		10 ppm (QE)		
X ¹³ CMS	intChoice	intb	maxo	Keep the same as MetTracer.
	enrichTol	0.1	0	To reduce false positives as recommended in their protocol
geoRge	ppm.s	6.5 ppm	25 ppm (QTOF) 10 ppm (QE)	Instrument platform dependent parameter. Keep the same as MetTracer.
EIMAVEN	Min signal baseline difference	0	3	Keep the same as MetTracer
	Minimum isotope-parent correlation	0.2	0.5	At least 50% frequency for the co-occurrence
	EIC extraction window	5 ppm	25 ppm (QTOF) 10 ppm (QE)	Instrument platform dependent parameter. Keep the same as MetTracer
	Match retention time	1.0 min	0.25 min	Keep the same as MetTracer.
	Minimum S/N ratio	1 sample	50% of samples	Keep the same as MetTracer.

Comment #2. *“After demonstrating the capabilities of MetTracer in a cell culture system, the manuscript presents application to Drosophila. The authors show that different groups of pathways incorporate labels at different rates, and that labeling follows a roughly exponential model, as can be calculated based on the output from MetTracer. Further, it is shown that head and muscle feature different labeling profiles and that, as Drosophila ages, there is a shift away from glycolysis toward serine and purine metabolism. These findings are well supported by the data, the results are in line with expectations based on previous work by the authors and others, and the coverage achieved with MetTracer is impressive.”*

Ans: We appreciate the positive comments from the Reviewer. We have also thoroughly revised the manuscript to address these comments. We feel that these comments have significantly strengthened the manuscript.

Comment #3. *“In addition, the authors claim that their data revealed disclose a “system wide loss of metabolic coordination that impacts both intra and inter tissue metabolic homeostasis significantly”. This statement appears to be based on the fact that Pearson correlation for different metabolites becomes much less significant for the 30-day-old cohort compared to the 3-day cohort. However, for this analysis, it is essential to consider that, per se, observation of aged animals results in increased heterogeneity, because individuals age at different rates (correspondingly, not all animals die at the same time). As far as I can see, this is not explicitly considered. Therefore, I do not think the parts of the manuscript discussing “loss of metabolic coordination” are supported by the data. As a result, the manuscript may not be appropriate for publication in its present form, though MetTracer appears to*

be a very useful tool for stable isotope labeling studies.”

Ans: Thanks a lot for the Reviewer's comment. We agree with the Reviewer and we share the same feeling with the fly community that **older flies have increased heterogeneity relative to young flies**. First of all, in our study, we chose the **30 days old flies** as aged representatives. **The median survival of flies is about 60 days and the mortality of 30 days old flies is only about 1.5%**. Second, to further reduce the heterogeneity effect, we pooled 15 flies to present as one biological sample, and a total of 10 biological independent samples (150 files in total) were used in one group (n=10) for metabolomics and isotope tracing analyses. Through the sample pooling, the heterogeneity from aged individuals is significantly reduced.

To prove the point, we calculated the coefficient of variation (CV) of labeling extent for each metabolite in 3d and 30d flies. The median CV values of 3d and 30d flies were 8.9% and 6.5%, respectively, which had no significant differences. Moreover, principal component analysis (PCA) of labeling extents of metabolites also demonstrated that the differences among biological replicates in either 3d or 30d group was relatively small. And the major metabolic differences stemmed from the age groups (3d vs. 30d). The related data has been added as **Supplementary Figure 16** in the revised manuscript.

In summary, by following the common practices in the fly community, we have demonstrated that the heterogeneity of 3d and 30d flies was similar, and the biological conclusions in this study were made from the effect of aging.

Supplementary Figure 16. Comparison of metabolic heterogeneity between 3d and 30d *Drosophila*. (a) Coefficient of variation (CV) values of labeled metabolites in 3d and 30d flies (n = 390; two-tailed Student's *t*-test). The lower, middle, and upper lines in box plots correspond to 25th, 50th, and 75th quartiles, and the whiskers extend to the most extreme data points within 1.5 interquartile ranges (IQR). (b) Principal component analysis (PCA) plot shows the separation of 3d and 30d flies (n = 10 in each group). Colored shadow, 95% confidence interval.

Minor points:

Comment #4. *“Figure 1: there are a few mistakes, e.g. “Correction for natural isotope...” and “isotopomer” in panel b.”*

Ans: Thanks for the Reviewer’s comment. We have carefully checked and corrected the spellings in the revised manuscript.

Comment #5. *“Line 125: explicitly state that this is the in-silico calculated number of isotopologues, of which then a subset is detected. This entire section is difficult to read and could benefit from clarification, especially with regard to what actually is done by MetTracer (the Methods section was somewhat helpful, but the main text section is nearly incomprehensible as is.”*

Ans: Thanks for the Reviewer’s comment. The entire workflow of MetTracer include three parts: generation of a targeted list for isotopologues; extraction of isotopologue peaks; isotopologue correction and quantification. In the first step, MetTracer calculated theoretical m/z values for all possible isotopologues (M0-Mn) for each metabolite (a total of 12,020 isotopologues for 1,347 metabolites). Second, MetTracer performed targeted extraction of all possible isotopologues from 1,347 metabolites in all samples. The results demonstrated that 10,663 isotopologues (88.7%) from 1,203 metabolites (89.3%) were successfully extracted in the labeled samples (**Supplementary Figure 2a and 2b**). Finally, MetTracer determined the labeling fraction for each isotopologue with the criterion of labeled fraction larger than 2% in more than 50% samples. As a result, MetTracer identified a total of 830 ^{13}C -labeled metabolites and 1,725 ^{13}C -labeled isotopologues which covered 66 metabolic pathways.

We have taken the reviewer’s comment and have clarified it as following in the revised manuscript.

*“Specifically, 1,347 metabolites were putatively annotated (215, 219, and 913 metabolites with MSI levels 1, 2 and 3, respectively). Then, the theoretical m/z values for 12,020 possible ^{13}C -isotopologues (M0-Mn) were calculated from the formulas of 1347 metabolites. MetTracer performed targeted extraction of all possible isotopologues and successfully extracted a total of 10,663 isotopologues (88.7%) from 1,203 metabolites (89.3%) (**Supplementary Figure 2a and 2b**), which ensured the high-coverage tracking of labeled metabolites. Finally, MetTracer determined the labeling fraction for each isotopologue with the criterion of labeled fraction larger than 2% in more than 50% samples. As a result, MetTracer identified a total of 830 ^{13}C -labeled metabolites and 1,725 ^{13}C -labeled isotopologues which covered 66 metabolic pathways (**Figure 1c and 1d**).”*

Comment #6: *“Line 127: It is unclear what is improved - recognition of peaks, or assignment as isotopologues of a specific metabolite.”*

“Line 139: Again, it’s unclear whether the better performance of MetTracer is due to better peak recognitions (i.e. fewer dropouts) when using MetTracer.”

Ans: Thanks a lot for the Reviewer's comment. The improvements of MetTracer have been well discussed in the first paragraph in the Discussion section. As being discussed, the main advancement of MetTracer comes from the integration of untargeted metabolite annotation and targeted isotopologue extraction. First of all, MetTracer benefited from the input of metabolite annotations of unlabeled samples from different bioinformatic tools, which ensures the high coverage of labeled metabolite extraction. Second, MetTracer has also made several data processing innovations to targeted isotopologue extraction, which ultimately contributes to the high coverage of labeled metabolites and high accuracy of isotopologue quantitation: 1) MetTracer performed targeted extraction of all possible isotopologues and employed the most intensive isotopologue peak instead of the M0 base peak for peak detection. For highly labeled metabolites wherein M0 peak has a relatively low intensity, the use of the most intensive isotopologue peak achieved a higher rate of successful peak detection; 2) The employment of targeted extracted ion chromatogram (i.e., target EIC in Figure 1b; see Methods) from unlabeled samples and calculation of peak-peak correlation (PPC) between isotopologue peak and target EIC peak also improved the accuracy and reduced the false positives; 3) The isotope contamination estimation and correction improved accurate quantitation of isotopologues.

In the revised manuscript, we further provided several metabolite examples (Supplementary Figures 3-6) to highlight the technical improvements of MetTracer. For example, we used Example 1 and Example 2 in Supplementary Figure 3 to demonstrate the advantages of targeted extraction compared with untargeted methods such as X¹³CMS and geoRge.

Example 1: peak M104T365 (M0 of serine) from Supplementary Figure 3a

Through targeted extraction, MetTracer correctly extracted all the isotopologues (M0-M3) of serine (m/z 104.0342 Da, 105.0376 Da, 106.0409 Da, and 107.0443 Da) in labeled 293T cell samples. As a comparison, X¹³CMS mistakenly assigned the peak (M104T365; m/z 104.0342 Da) as the M3 isotopologue for another peak (m/z 101.0247 Da, RT 364.1s). The possible reason for this wrong assignment is because X¹³CMS performs the untargeted discovery of the base peak M0 in a defined retention time window. If there were several metabolites co-eluted, mistakes may happen as shown in Example 1.

Example 2: peak M181T288_2 (M0 of threitol) from Supplementary Figure 3b

Through targeted extraction, MetTracer correctly extracted all isotopologues M0-M4 of threitol (m/z 181.0707 Da, 182.0740 Da, 183.0774 Da, 184.0807 Da, and 185.0841 Da) in labeled 293T cell samples. As a comparison, geoRge mistakenly assigned the peak (M181T288_2; m/z 181.0707 Da) as the M1 isotopologue peak for another peak (m/z 180.0607 Da, RT 287.4s, annotated as tyrosine). The reason for this wrong assignment is similar to X¹³CMS. geoRge performs untargeted discovery of the base peak M0 and is prone to errors if there were several metabolites co-eluted.

Supplementary Figure 3. Examples to demonstrate the advantages of targeted extraction in MetTracer. (a) Left panel, extracted ion chromatograms of isotopologue peaks of serine; middle panel, output results by MetTracer; right panel, output results of the isotopologue peak group containing M104T365 by X¹³CMS. (b) Left panel, extracted ion chromatograms of isotopologue peaks of threitol; middle panel, output results by MetTracer; right panel, output results of the isotopologue peak group containing M181T288_2 by geoRge. The examples were from labeled 293T cell samples acquired with the TripleTOF 6600 instrument.

We use Example 3 and Example 4 in Supplementary Figure 4 to demonstrate highly labeled metabolites can be extracted in MetTracer but failed in EI-MAVEN.

Examples 3 and 4: peak M221T224 and M170T331 from Supplementary Figure 4

In the example of peak M221T224 (M0 of 4-Fumarylacetoacetate; **Supplementary Figure 4a**), the metabolite was highly labeled in M6 and M8, and the base peak M0 had a relatively low intensity (labeling fraction, 0.03). In MetTracer, the highest isotopologue peak M8 instead of M0 peak was used for peak detection. Therefore, all the isotopologues M0-M8 of 4-Fumarylacetoacetate (m/z 221.0057 Da, 222.0090 Da, 223.0124 Da, 224.0157 Da, 225.0191 Da, 226.0224 Da, 227.0258 Da, 228.0291 Da, 229.0325 Da) were successfully detected in labeled samples. A similar example for peak M170T331 (2-Amino-6-oxoheptanedioate) was also provided in **Supplementary Figure 4b**. In this example, the highest isotopologue peak M3 instead of M0 peak was used for peak detection.

As a comparison, in EI-MAVEN, each isotopologue peak need to compared with M0 base peak for peak shape similarity. For highly labeled metabolites when the base peak M0 had a low intensity, recognition of other isotopologue peaks often failed. Due to this reason, both M221T224 and M170T331 had no results in EI-MAVEN analysis.

Supplementary Figure 4. Examples to demonstrate the performance of MetTracer for highly labeled metabolites. (a) Left panel, extracted ion chromatograms of isotopologue peaks of 4-Fumarylacetoacetate; right panel, output results by MetTracer. (b) Left panel, extracted ion chromatograms of isotopologue peaks of 2-Amino-6-oxoheptanedioate; right panel, output results by MetTracer. The examples were from labeled 293T cell samples acquired with the TripleTOF 6600 instrument.

Supplementary Figure 5. Example to demonstrate the utilization of PPC score for correct peak grouping in MetTracer. (a) Targeted EIC of M153T266 recorded in the targeted list. (b) Isotopologue

peaks extracted by MetTracer and their calculated PPC scores. **(c)** Extracted ion chromatograms of isotopologue peaks of oxalureate finally decided by MetTracer. **(d)** Output results of oxalureate by MetTracer. The examples were from labeled 293T cell samples acquired with the TripleTOF 6600 instrument.

We use Example 5 in **Supplementary Figure 5** to demonstrate the use of PPC score for correct peak grouping in MetTracer.

Example 5: peak M153T266 (M0 of oxalureate) from Supplementary Figure 5

In the example of peak M153T266 (oxalureate), the targeted EIC was first recorded. MetTracer extracted the isotopologue peaks (M0-M3) according to m/z values (m/z 152.9907 Da, 153.994 Da, 154.9974 Da, 156.0007 Da) and retention times, and calculated their PPC scores with targeted EICs. Isotopologue interferences were removed due to their PPC scores lower than 0.6. Then, the peak range of M0 was used for quantification.

We use Example 6 in **Supplementary Figure 6** to demonstrate the performance of isotope contamination correction in MetTracer.

Example 6: peak M114T366 (M0 of Iminodiacetate) from Supplementary Figure 6

For the example of M114T366 (iminodiacetate), in unlabeled samples, the M3 isotopologue peak (m/z 117.0286 Da) was contaminated by other co-eluting metabolites, and its labeled fraction was 0.12. After contamination correction, the labeled fraction was determined as 0. This result showed that contamination correction contributes to accurate quantification of the isotopologue peaks.

Supplementary Figure 6. Example to demonstrate the isotope contamination correction in MetTracer. **(a)** Extracted ion chromatograms of isotopologue peaks of iminodiacetate before contamination correction in unlabeled samples. **(b)** Labeling extent result of iminodiacetate before contamination correction. **(c)** Labeling extent result of iminodiacetate after contamination correction. The examples were from unlabeled 293T cell samples acquired with the TripleTOF 6600 instrument.

Comment #7. *“Line 133: it is unclear what “good quantitative consistency” means, relative to the statement in the previous sentence.”*

Ans: Thanks a lot for the Reviewer’s comment. We have evaluated the quantitation consistency at both metabolite level and isotopologue level. Labeling extents (LE) of 830 metabolites were calculated using MetTracer and compared to the results obtained from manual analysis using Skyline (**Figure 1e**). Similarly, in **Supplementary Figure 2c**, we compared the peak intensities of 1725 isotopologues obtained from MetTracer to those from manual analysis using Skyline. Metabolites with relative errors less than 30% were considered as good quantitative consistency. Relative error was calculated as:

$$\text{Relative error} = \frac{LE(\text{MetTracer}) - LE(\text{Skyline})}{LE(\text{Skyline})}$$

In metabolomics community, the relative error less than 30% is considered as reliable quantitation. Therefore, we used 30% as the cut-off for quantification accuracy.

Comment #8. *“Line 168: “was the fastest labeling group” - isn't this simply the result of the criteria used for clustering?”*

Ans: Thanks for the Reviewer’s comment. The labeling extents at 6 time points for each metabolite were used for hierarchical clustering analysis (HCA). Indeed, the results clearly showed that cluster 1 was the fastest labeling group. However, HCA not only divided the metabolites into three groups, but also determined which metabolites belonged to cluster 1, cluster 2 or cluster 3. Then, the related enriched metabolic pathways could be disclosed by these metabolites.

We have clarified the points in the revised manuscript, and revised the related statement as:

“The fastest labeling cluster 1 reached the isotopic steady state within 3-6 h. Pathway enrichment analysis revealed that labeled metabolites in cluster 1 were mainly in fructose and mannose metabolism and galactose metabolism”

Comment #9. *“Line 185: “Further, ...” This again merely restates that labeling rates were found to be different for different metabolite groups, i.e. the different clusters correspond to metabolites from different pathways.”*

Ans: Thanks a lot for the reviewer’s comment. To clarify, we quantitated the mean labeling rates of 19 metabolic pathways using the mean labeling rates of metabolites in each pathway. We aimed to reveal metabolic kinetics at the pathway level. We have clarified the points in the revised manuscript, and revised the related statement as:

“We further investigated the mean metabolic labeling rates for metabolites from 19 metabolic pathways in Drosophila head, and provided an estimation of metabolic kinetics on the pathway level.”

Reviewer #3

Remarks to the Author: “Wang et al. describe MetTracer, a novel system for tracing metabolites across flies of different ages to gain insights into how head and muscle metabolomes alter with age. The analyses utilized for this study provide clear results to demonstrate the utility of MetTracer. There are a number of concerns that must be addressed for this paper to be publishable.”

Ans: The authors would like to thank the Reviewer for the helpful comments. We have thoroughly revised the manuscript to address these comments. We feel these comments have significantly strengthened the manuscript.

Comment #1. “The study used a w¹¹¹⁸ fly model. While this strain is commonly used as a ‘wild type’ of sorts, it notably has a mutation to a gene that influences purine transport, neurotransmitter function, and eye function. There is a strong likelihood that this mutation alters the metabolome, thus skewing the results presented in this study. While this does not hurt the main point the authors are making when demonstrating the utility of MetTracer, it does provide unintended context to the dataset they generated, especially considering how strongly purine metabolism was represented in the dataset. It would be worth discussing this and adding appropriate references.”

Ans: Thanks for the Reviewer’s comment. In this study, we used MetTracer to study the effect of aging on the metabolome of 3d and 30d flies, and both of them have the same genotype w¹¹¹⁸. We have read the related literatures and agree with the Reviewer that, although the w¹¹¹⁸ flies are widely used as wild-type controls in fly studies, it is worth noting that w mutant had an impact on fly metabolism. Specifically, the *Drosophila w* gene encodes an ATP binding cassette transporter, which contributes to transportation of metabolites such as guanine, tryptophan and kynurenine (*J. Mol. Biol.*, 1984, [https://doi.org/10.1016/0022-2836\(84\)90021-4](https://doi.org/10.1016/0022-2836(84)90021-4); *Nucleic Acids Res.*, 1990, <https://doi.org/10.1093/nar/18.6.1633>; *Biochem. Genet.*, 1979, <https://doi.org/10.1007/BF00498891>; *Biochem. Genet.*, 1975, <https://doi.org/10.1007/BF00484918>). The w mutant flies were also observed with low levels of the biogenic amines, such as serotonin, dopamine, and histamine (*J. Exp. Bio.*, 2008, <http://doi.org/10.1242/jeb.021162>; *Proc. Natl. Acad. Sci.*, 2008, <https://doi.org/10.1073/pnas.0710168105>). *Drosophila w* gene is expressed principally in eyes and excretory organs and testes, but very low levels are observed in the glia and neurons of the brain and various other tissues (*Genetica*, 2000, <https://doi.org/10.1023/A:1004115718597>; *Nat. Genet.*, 2007, <https://doi.org/10.1038/ng2049>). In the revised manuscript, we have taken the Reviewer’s suggestion and added the following discussion:

“In this study, we used w¹¹¹⁸ fly model to demonstrate the utility of MetTracer in studying aging, and we considered that, the metabolic changes between 3d flies and 30d flies were caused by aging here. Although the w¹¹¹⁸ flies were widely used as wild-type controls in many fly aging studies, it is worth noting that, previous studies have discovered that the w mutant of w¹¹¹⁸ flies have an impact on

fly metabolism. The *Drosophila w* gene encodes an ATP binding cassette transporter, which contributes to transportation of metabolites such as guanine, tryptophan and kynurenine. The *w* mutant flies were also observed with low levels of the biogenic amines, such as serotonin, dopamine, and histamine. The *Drosophila w* gene is expressed principally in eyes and excretory organs and testes, but has very low levels observed in the glia and neurons of the brain and various other tissues. Although whether the metabolome of *w*¹¹¹⁸ will be affected by aging is unclear, we should still be cautious that this mutation alters the metabolism of *Drosophila*.”

Comment #2. “In Figure 1f, I wonder why the comparison in reproducibility and false positives is restricted to just being between MetTracer and EI-MAVEN. Considering EI-MAVEN was the tool which had the least overlap with MetTracer (presented in Figure 1d), I think it would be useful to compare its reproducibility to other tools which have more similar overlap in signatures (or better yet, a comparison between all tools presented in 1d).”

Ans: Thanks a lot for the Reviewer’s comment. We have taken the Reviewer’s suggestion and added the reproducibility comparison of four software tools in the revised **Figure 1f** (for the QTOF dataset) and **Supplementary Figure 7g** (for the Orbitrap dataset).

Figure 1f. Relative standard deviation (RSD) distributions of metabolites and isotopologues obtained from MetTracer and other software tools (n=6 technical replicates of 293T cell samples). The black dots represent median RSD. The data was acquired with the TripleTOF 6600 instrument.

Supplementary Figure 7g. Relative standard deviation (RSD) distributions of metabolites and isotopologues obtained from MetTracer and other software tools (n=6 technical replicates of 293T cell samples). The black dots represent median RSD. The data was acquired with the Exploris480 instrument.

The related statements have also been revised as:

“Median RSDs of labeled fractions for metabolites and isotopologues obtained using MetTracer were 4.9% and 23.1%, respectively, which were close to the median RSDs obtained using geoRge and X¹³CMS. As a comparison, EI-MAVEN resulted in significantly higher median RSD values of 77.6% and 121.7% for the labeled metabolites and isotopologues, respectively (Figure 1f)”

For comparison of false positive rates (FPR), there is a technical limitation that we can only calculate FPR for MetTracer and EI-MAVEN. Specifically, unlabeled samples (instead of labeled samples) were analyzed using MetTracer and EI-MAVEN. **The false positives were defined as the labeled isotopologues and metabolites in unlabeled samples.** Both MetTracer and EI-MAVEN could support the FPR analysis (see Methods for details). However, X¹³MS and geoRge require **both unlabeled and labeled samples as inputs** to determine the labeled isotopologues and metabolites. Therefore, unlabeled samples were necessary in the data processing and served as control samples. As a result, it is not possible to estimate the false positive rates of X¹³MS and geoRge using the similar method as MetTracer and EI-MAVEN. We have indicated this limitation in the Methods of the revised manuscript as follows:

“There is a technical limitation to estimate false positive rate in untargeted methods such as X¹³CMS and geoRge. They require both unlabeled and labeled samples as inputs to determine the labeled isotopologues and metabolites. The unlabeled samples were necessary in the data processing and served as control samples. As a result, it is not possible to estimate the false positive rates of X¹³CMS and geoRge using the similar method as MetTracer and EI-MAVEN.”

Comment #3. *“While the tool described is novel, age-related metabolomic analyses in Drosophila have been performed many times (Coquin et al 2008, Avanesov et al 2014, Hoffman et al 2014, Laye et al 2014, Zhou et al 2017, Jin et al 2020, Parkhitko et al 2020, Tapia et al 2021, Hall et al 2021, Zhao et al, 2022). A comparison between the results from these studies and the new approach presented in this manuscript would help demonstrate the value of the authors’ method. Depending on the fly strains used in the other studies, this could also alleviate the concerns about the w1118 model or affirm the results from this study.”*

Ans: Thanks a lot for the reviewer’s comment. We have reviewed the papers mentioned above and have taken the comments by revising the manuscript thoroughly. In the revised manuscript, we have summarized the fly strains, metabolic observations and comparison of our study with these studies in **Supplementary Table 7** and added the following discussion:

“Drosophila is a widely used model organism to study aging. Previous studies using traditional metabolomics methods have disclosed metabolic changes associated with aging in varied genotypes of Drosophila (Supplementary Table 7). For example, Chang et. al., Hoffman et.al. and our group reported the declination of metabolites in glycolysis pathway during aging. Metabolites in amino acid

metabolism were found to have correlation with aging. For example, Tapia et.al. revealed that reduction in amino acids such as histidine, tyrosine, and leucine is protective against aging. Instead, Parkhitko et.al. showed that metabolic intervention with supplementation of tyrosine contributed to lifespan extension. In addition, Laye et.al. reported that S-adenosyl-methionine (SAM) cycle was downregulated in flies under diet restricted condition compared to the age-matched flies with normal diet, suggesting that the long-lived flies have lower level of SAM. Similarly, enhancement of SAM catabolism by overexpressing glycine N-methyltransferase (Gnmt) has also been demonstrated to extend lifespan of *Drosophila*, as reported by Obata et.al. Elevated purine metabolism was also found to be associated with short lifespan of *Drosophila*.

Despite the accumulative aging-related metabolites and metabolic pathways being uncovered by measuring the changes of metabolite concentrations associated with aging, these were independently studied, and the underlying mechanism by which coordination of metabolic activities impacts aging is not elucidated due to the lack of appropriate technologies. Here, in addition to metabolite concentrations, we demonstrated that the global isotope tracing metabolomics can provide an orthogonal perspective to the changed metabolome by monitoring the labeling parameters at different levels, thereby supporting quantitative comparison of metabolic activities during aging. We discerned the age-dependent changes of metabolic activities in *Drosophila*, in which carbohydrate related metabolic pathways were enriched with metabolites of declined metabolic activities in aging, while amino acid related metabolic pathways and purine metabolism were enriched with metabolites of enhanced metabolic activities in aging, which were consistent with the observations in previous studies (**Supplementary Table 7**). This suggests that distinct metabolite utilization in different biological scenarios can explain the age-dependent metabolic responses in physiological status. A key contribution of this work is to reveal a system-wide loss of metabolic coordination during aging in *Drosophila* using global isotope tracing metabolomics. This uniquely system-wide loss was evidenced by the massive reduction of metabolite-metabolite correlations within the head and muscle tissues and across tissues. We showed that glycolysis was skewed to one-carbon metabolism.....”

Supplementary Table 7 Comparison of metabolic observations between our study and other studies

Fly strain	Major metabolic conclusions	Reference	Related conclusions in MetTracer
Oregon-R wild type	In the hypoxia model of Drosophila , old flies showed lower TCA cycle fluxes during recovery from hypoxia.	Coquin et al., 2008, Molecular Systems Biology [ref.43]	No related observations in the context of hypoxia.
Canton-S wild type	The number of detected metabolites increased as a function of age, followed by cessation of the increase in late life.	Avanesov et.al., 2014, eLife [ref. 44]	Aging influenced the system-wide metabolic coordination.
15 inbred fly lines from DGRP	Glycolysis related metabolites were declined with aging. Carnitines were declined with aging. Metabolites associated with glycerophospholipids showed either increases or declines with age.	Hoffman et.al., 2014, Aging Cell [ref. 45]	Consistent with the literature. Glycolysis was down-regulated with aging. No related observations in glycerophospholipids and carnitines.
w1118	Diet restriction (DR) significantly slowed age-related changes in the metabolome and enhanced metabolite correlations with age in Drosophila . Amino acids such as leucine, tryptophan and betaine showed declines upon DR, and increased in aging. Beta-alanine biosynthesis, SAM cycle and methionine degradation were downregulated upon DR, and increased in aging.	Laye et.al., 2015, Aging Cell [ref. 46]	Metabolite-metabolite correlations were lost during normal aging. Consistent with the literature. Betaine was increased with aging. SAM and SAH were increased with aging. Metabolite precursors of one-carbon metabolism such as serine and glycine were increased with aging.
w1118, yw, Canton-S	Enhancement of SAM catabolism extends the lifespan in Drosophila .	Fumiaki Obata, Masayuki Miura, 2015 Nature Communications [ref. 35]	Consistent with the literature. SAM and SAH were increased with aging. Metabolite precursors of one-carbon metabolism such as serine and glycine were increased with aging.
w1118	Metabolites such as glutamine, alanine, glycine, tyrosine, tryptophan were highly correlated with the increase of age. Metabolites including leucine, valine, methionine were correlated with sex in aging process.	Zhou et.al., 2017, Experimental Gerontology [ref.47]	Metabolites such as glutamine, alanine and glycine were altered with aging.
w1118	Glycolysis was downregulated with aging, and the	Ma et.al., 2018,	Consistent with the literature. Glycolysis was

	activity of glycolysis was enhanced in long-lived PRC2 mutants.	eLife [ref. 41]	declined with aging.
B3, O1, O3	Level of tyrosine was increased with age in long-lived flies. Tyrosine supplementation and downregulation of enzymes in the tyrosine degradation pathway significantly extend Drosophila lifespan.	Parkhitko et.al., 2020, eLife [ref. 48]	No related observations.
178 inbred fly lines from DGPR	Proteinogenic amino acids and metabolites involved in α -KG and glutamine metabolism were correlated with the magnitude of the lifespan response upon diet restriction.	Jin et.al., 2020, PLOS Genetics [ref. 49]	α -KG, glutamine, and proteinogenic amino acids such as proline, alanine were associated with aging.
w ^{Dah}	High-sugar feeding shortened fly lifespan through increasing purine catabolism.	Dam et.al., 2020, Cell Metabolism [ref. 40]	Purine metabolism was observed to be enhanced in old flies.
w ^{Dah} , w ^{iso31} , Canton-S	Purine metabolism was increased in aged flies, and purine metabolites were impacted by gut bacterial species distinctively.	Yamauchi et.al., 2020, iScience [ref. 50]	Consistent with the literature. Purine metabolism was observed to be enhanced in old flies.
Oregon-R	TCA cycle components were decreased with aging. Triacylglycerols and phospholipids were increased with aging. Amino acids such as histidine, tyrosine and leucine have trends towards lower levels with aging.	Tapia et.al., 2021, International Journal of Molecular Sciences [ref. 52]	TCA cycle components such as α -KG and succinate were decreased with aging. No related observation for triacylglycerols, phospholipids, histidine, tyrosine and leucine
w1118	Decreases in riboflavin and metabolites in purine metabolism were observed in the eyes of aged flies, which were distinct from those observed in the head or whole fly. Metabolites involved in SAM regeneration was increased in the eyes of aged flies.	Hall et.al., 2021, Molecular Cell Proteomics [ref.51]	Purine metabolism was observed to be upregulated during aging in fly head. SAM and SAH were increased with aging.
20 fly lines from DGPR	Metabolites such as proline, glutamate, glutamine, and tryptophan were associated with aging.	Zhao et.al., 2021, Aging Cell [ref. 53]	Metabolites such as proline, glutamine and glutamate were dysregulated during aging.

Minor concerns:

Comment #4. *“There are a number of typographical errors:*

- *In line 35, ‘facilitates’ is spelled incorrectly.*
- *Line 155 has an unnecessary space before the comma.*
- *The sentence in lines 227-229 is incomplete and probably should be combined with the preceding sentence.*
- *Labeling of ages is not consistent throughout the manuscript (e.g. ‘30 d’ in line 324 vs ‘30-d’ in line 328 vs ‘30d’ in the figures)*
- *In line 388, ‘age-dependent’ is misspelled.*
- *In line 442, ‘wild-type’ is misspelled as ‘wide type.’”*

Ans: Thanks for the Reviewer’s comment. We have carefully checked and corrected the spellings in the revised manuscript and Supplementary Information.

Comment #5. *“Why were head and muscle chosen as the tissues of interest?”*

Ans: Thanks for the Reviewer’s comment. In our previous study, we have identified important metabolic changes during aging in head tissue and muscle tissue ([eLife](https://doi.org/10.7554/eLife.35368.001), 2018, <https://doi.org/10.7554/eLife.35368.001>). In addition, both tissues are relatively easy to obtain without sophisticated dissection.

Comment #6. *“Figure 4 requires some additional descriptive details. How were the metabolites within each cluster in the Circos plots arranged? And how are inter-cluster correlations affected with age?”*

Ans: Thanks a lot for the Reviewer’s comment. In **Figure 4**, we showed the numbers of significant metabolite correlations for intra- and inter- clusters. The colored circles refer to clusters. Ribbons connecting metabolite clusters refer to the significant correlations between metabolites. Ribbon thickness refers to the number of significant metabolite-to-metabolite correlations. Metabolites are arranged by labeling rate clusters, and there is no specific arrangement of metabolites in each cluster.

To clarify, in the revised manuscript, we have added the details in figure caption as following: *“The colored circle refers to the labeling rate cluster. Ribbon connecting metabolite clusters refers to the significant correlations between metabolites. Ribbon thickness refers to the number of significant metabolite-to-metabolite correlations.”*

We have also added the details in the Method section as following: *“The labeling extent values after 24 h labeling were used for Pearson correlation calculation between metabolites from each cluster. Only metabolite correlations with P value less than 0.05 after FDR correction were retained to construct the correlation network. The correlation network was visualized by R package “circlize”*

(version 0.2.10).”

According to the comment of Reviewer, we have also added the statistics of inter-cluster correlations affected by age in the revised **Supplementary Figure 13**. The related description has been added in main text: “We also investigated the effect of age on inter-cluster correlations. In fly head and muscle, the correlations between metabolites in fast-labeling cluster 1 and other clusters were severely affected by aging, wherein about 60% correlations were diminished. The correlations between metabolites in cluster 2 and cluster 3 were slightly affected by aging within head and muscle tissues. The inter-cluster metabolic coordination between head and muscle was also impacted by aging, especially metabolites in cluster 3 with other clusters, wherein more than 60% of correlations were diminished.”

Supplementary Figure 13. Alternations of inter-cluster metabolic correlations during aging. Bar plot showing the numbers of inter-cluster metabolite-metabolite correlations in young and old *Drosophila*: head tissue (a), muscle tissue (b), inter tissue (c).

Comment #7. “The individual data points in some of the box plots are too small to make out, especially in Figure 5d. If the size of these could be increased, it would be helpful.”

Ans: Thanks a lot for the reviewer’s comment. In the revised manuscript, we have increased the size of data points in Figure 5d.

Comment #8. “How was gene expression determined in Figure 5? This should be briefly mentioned in the text and detailed in the Methods.”

Ans: We appreciate the Reviewer’s comment. The gene expression data were obtained from RNA-Seq experiment. We have added the experimental description in the Methods as the following. The RNA-Seq results have also been provided in **Supplementary Data 9**.

“PolyA-selected RNA-seq. For RNA-seq experiments, dissected fly head tissues (3d and 30d) were used. Tissues were homogenized in a 1.5 mL tube containing 1 mL of Trizol Reagent (Thermo Fisher Scientific, USA). RNA isolation was followed in accordance with manufacturer’s instruction.

RNA was resuspended in DEPC-treated RNase-free water (Thermo Fisher Scientific). TURBO DNA free kit was used to remove residual DNA contamination according to manufacturer's instruction (Thermo Fisher Scientific). 1 µg of total RNA was used for sequencing library preparation. PolyA-tailed RNAs were selected by NEBNext Poly(A) mRNA Magnetic Isolation Module (New England Biolabs, USA), followed by the library preparation using NEBNext Ultra RNA library Prep Kit for Illumina according to manufacturer's instruction (New England Biolabs, USA). Libraries were pooled and sequenced on the Illumina Miseq platform with single end 100 bps (Illumina, USA). Sequencing reads were mapped to the reference genome dm6 with STAR2.3.0e by default parameter. The read counts for each gene were calculated by HTSeq-0.5.4e htseq-count with parameters '-m intersection-strict -s no' with STAR generated SAM files. The count files were used as input to R package "DESeq" for normalization."

Comment #9. "Various points in the manuscript (like lines 265 and 335) mention a "previous report" but do not provide a citation."

Ans: We appreciate the Reviewer's comment. We have carefully checked them and added the related citations in the revised manuscript.

Comment #10. "For the described PRC2 mutation, is this referring to mutations to genes in the Polycomb repressive complex 2? This is needs to be clarified."

Ans: Thanks a lot for the reviewer's comment. Yes, the described PRC2 mutation is referred to mutations to genes in the Polycomb repressive complex 2. The detailed genotype is *Pcl*^{c421/+}; *Su(z)12*^{c253/+}. We have clarified the genotype in the Methods of the revised manuscript.

Comment #11. "Reference to Yuan et al. in the discussion (line 376) requires a citation."

Ans: Thanks for the reviewer's comment. We have added the citation in the revised manuscript.

Comment #12: "Some of the Discussion section is redundant (e.g. lines 396 and 409 repeat each other)."

Ans: Thanks for the reviewer's comment. The sentence in line 396 "glycolysis shifted to one-carbon metabolism that further ..." has been removed in the revised manuscript.

Comment #13: "Figure 4/results- typo in subplot c legend, liner regression should be linear regression"

Ans: Thanks for the reviewer's comment. We have corrected the typo errors in the revised manuscript.

Comment #14: *“What is purpose of 5a? What are the non-labeled nodes?”*

Ans: We appreciate the Reviewer’s comment. To clarify, we used the metabolite pyruvate as an example to demonstrate the loss of metabolic coordination between metabolites during aging. In Figure 5a, each node (labeled and non-labeled) represents a metabolite (e.g., serine) that has a metabolic correlation with pyruvate in 3d flies, and no correlation with pyruvate in 30d flies. We have also clarified this in the revised manuscript, and provided the list of metabolites and their correlation with pyruvate in **Supplementary Table 6**.

*“In young Drosophila head tissues, after the 24-h labeling, pyruvate had metabolic correlations with other 23 metabolites such as serine, succinate and glucose (**Supplementary Table 6**). Interestingly, these metabolic correlations with pyruvate were completely disrupted in old Drosophila”.*

Comment #15: *“Same with 5f, a model, but only thing that changes is thickness of roundabout and the seesaw.”*

Ans: Thanks a lot for the Reviewer’s comment. We have revised the Figure 5f to make it more intuitive to read.

REVIEWERS' COMMENTS

Reviewer #1 (Remarks to the Author):

The authors have addressed my concerns

Reviewer #3 (Remarks to the Author):

I want to congratulate the authors on an excellent job to respond to the reviewer's comments.

Reviewer #1:

Remarks to the Author: *"The authors have addressed my concerns"*

Ans: We thank the positive comments from the reviewer.

Reviewer #3:

Remarks to the Author: *"I want to congratulate the authors on an excellent job to respond to the reviewer's comments."*

Ans: We thank the positive comments from the reviewer.